# MUSE: Model-Agnostic Tabular Watermarking via Multi-Sample Selection

**Liancheng Fang[1], Aiwei Liu[2,†], Henry Peng Zou[1], Yankai Chen[3,4], Hengrui Zhang[1], Zhongfen Deng[1,‡], Philip S. Yu[1]**

[1] University of Illinois Chicago, [2] Tsinghua University, [3] MBZUAI, [4] McGill University

## Abstract

We introduce MUSE, a novel watermarking paradigm for tabular generative models. Existing approaches often exploit DDIM invertibility to watermark tabular diffusion models, but tabular diffusion models suffer from poor invertibility, leading to degraded performance. To overcome this limitation, we leverage the computational efficiency of tabular generative models and propose a multi-sample selection paradigm, where watermarks are embedded by generating multiple candidate samples and selecting one according to a specialized scoring function. The key advantages of MUSE include (1) **Model-agnostic**: compatible with any tabular generative model that supports repeated sampling; (2) **Flexible**: offers flexible designs to navigate the trade-off between generation quality, detectability, and robustness; (3) **Calibratable**: theoretical analysis provides principled calibration of watermarking strength, ensuring minimal distortion to the original data distribution. Extensive experiments on four datasets demonstrate that MUSE substantially outperforms existing methods. Notably, it reduces the distortion rates by $84 - 88\%$ for fidelity metrics compared with the best performing baselines, while achieving $1.0$ TPR@0.1%FPR detection rate. Code is publicly available at this URL.

## 1 Introduction

The rapid development of tabular generative models (Kotelnikov et al., 2023; Gulati and Roysdon, 2024; Castellon et al., 2023; Zhang et al., 2024c; Shi et al., 2024; Fang et al., 2025; Zhang et al., 2025) has significantly advanced synthetic data generation capabilities for structured information. These breakthroughs have enabled the creation of high-quality synthetic tables for applications in privacy preservation, data augmentation, and missing value imputation (Zhang et al., 2024b; Hernandez et al., 2022; Fonseca and Bacao, 2023; Assefa et al., 2020). However, this advancement concurrently raises serious concerns about potential misuse, including data poisoning (Padhi et al., 2021) and financial fraud (Cartella et al., 2021). To address these risks, watermarking has emerged as a pivotal technique. By embedding imperceptible yet robust signatures into synthetic data, watermarking facilitates traceability, ownership verification, and misuse detection (Liu et al., 2024b).

Earlier works on tabular data watermarking utilize *edit-based watermarking* (Zheng et al., 2024; He et al., 2024), embedding signals by modifying table values. However, this approach has a fundamental limitation with tabular data: direct value alterations, especially in columns with discrete or categorical data, can easily corrupt information or render entries invalid. For instance, such edits might introduce non-existent categories (Gu et al., 2024; Lin et al., 2021) or push values across critical decision boundaries (Ngo et al., 2024), significantly compromising data integrity. Recently, *generative watermarking* has emerged as an alternative approach for tabular data, drawing from successful techniques in diffusion models for images and videos (Wu et al., 2025; Yang et al., 2024; Wen et al., 2023; Hu et al., 2025). This approach leverages the reversibility of DDIM samplers (Song et al., 2020a) by initializing generation with patterned Gaussian noise and, during watermark detection, assessing its correlation with the noise reconstructed through the inverse process. TabWak (Zhu et al., 2025) applies this concept to tabular diffusion models (Zhang et al., 2024c; Kotelnikov et al., 2023; Lee et al., 2023; Kim et al., 2022). Unlike edit-based watermarking, generative watermarking maintains better generation quality since the watermark is embedded within noise patterns that closely resemble Gaussian distributions, minimizing impact on the generated content.

---

†Corresponding author. ‡Work done prior to Amazon.

However, watermarking tabular diffusion models is significantly more challenging than for image and video diffusion models. This stems from the **substantially lower accuracy of DDIM inverse processes** in tabular diffusion models, as shown in Figure 1 (left). When using the same Gaussian shading algorithm (Yang et al., 2024), tabular modality exhibits the lowest reversibility accuracy. This challenge arises because tabular diffusion models incorporate multiple additional algorithmic components that are difficult to reverse, such as quantile normalization (Amaratunga and Cabrera, 2001) and Variational Autoencoders (VAEs) (Kingma and Welling, 2013) used in TabSyn (Zhang et al., 2024c). During watermark detection, the entire data processing pipeline must be inverted to recover the watermark signal, but this process accumulates errors

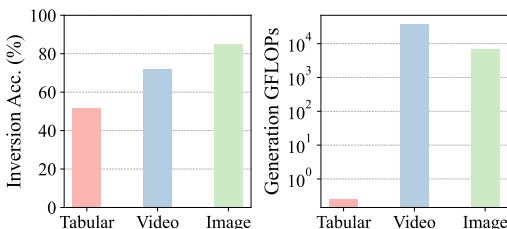

Figure 1: **Left**: Tabular diffusion models exhibit the lowest inversion accuracy (bit accuracy) when compared to video and image diffusion models. **Right**: Tabular diffusion models require much fewer generation GFLOPs than video and image diffusion models. Models used: TabSyn (Zhang et al., 2024c) (tabular), Stable Diffusion (Rombach et al., 2022; Blattmann et al., 2023) (image/video).

as precisely reversing each step is often difficult or impossible. Key challenges in the inversion process include: **(1)** inverting quantile normalization is inherently problematic as this transformation is non-injective; **(2)** VAE decoder inversion relies on optimization methods without guarantees of perfect implementation. Due to limitations in tabular DDIM inversion accuracy, watermark detectability becomes highly dependent on model implementation, severely restricting its application scope and practical utility (see Section D for more details).

This paper introduces MUSE, a model-agnostic watermarking paradigm for tabular data that operates without relying on the invertibility of diffusion models. A key insight enabling our approach is that tabular data generation demands **significantly less computation** than image or video generation, as shown in Figure 1 (right). This computational efficiency makes a multi-sample selection process practical: MUSE leverages this by generating multiple candidate samples for each data row and embedding the watermark by selecting one candidate based on a keyed watermark scoring function, which is calculated using values from specific columns. We present MUSE as a general paradigm and introduce two specific implementations that navigate the crucial trade-off between data fidelity and watermark detectability/robustness: **(1)** Joint-Vector (JV) hashing, tailored for minimal distortion (distribution-preserving), and **(2)** Per-Column (PC) hashing, designed for maximal robustness and detectability. We ground this paradigm in rigorous theoretical analysis, providing a precise method to calibrate detectability and establishing conditions for distortion-free watermarking. Validated across diverse datasets, MUSE demonstrates high watermark detectability and strong robustness against attacks while maintaining the underlying model's generation quality.

**Our Contributions.** We summarize the main contributions of this paper as follows:

- We propose tabular watermarking via multi-sample selection (MUSE), a novel generative watermarking paradigm for tabular data that completely avoids the inversion of generative and data processing pipelines, ensuring broad compatibility with any tabular generative model.

- We demonstrate the flexibility of the MUSE paradigm, showing how different score function designs enable a controllable trade-off between generation quality, detectability, and robustness.

- We provide theoretical analysis of MUSE, establishing its detectability for precise strength calibration and identifying the conditions for achieving distribution-preserving watermarking.

- Extensive experiments across multiple tabular datasets validate MUSE's superior performance in generation quality, watermark detectability, and robustness against various tabular-specific attacks.

## 2  PRELIMINARIES

**Tabular Generative Models.**  A tabular dataset with $N$ rows and $M$ columns consists of $i.i.d.$ samples $(\mathbf{x}_i)_{i=1}^N$ drawn from an unknown joint distribution $p_{\text{data}}(\mathbf{x})$, where each $\mathbf{x}_i \in \mathbb{R}^M$ (or mixed-type space) represents a data row with $M$ features. A tabular generative model aims to learn a parameterized distribution $p_\theta(\mathbf{x}) \approx p_{\text{data}}(\mathbf{x})$ to generate new realistic samples.

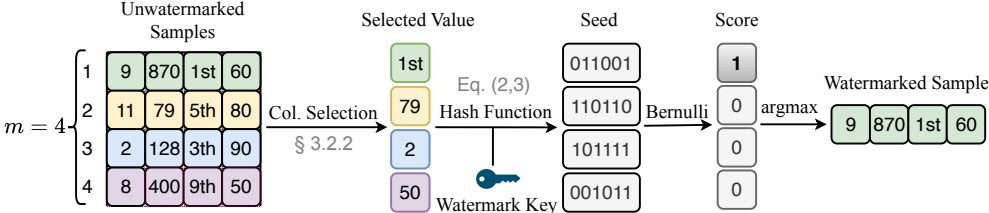

Figure 2: An overview of the MUSE watermark generation process. MUSE operates by generating multiple samples and selecting the highest-scoring sample (ties are broken randomly). The selected row is appended to the watermarked table, while others are discarded.

**Watermark for Tabular Generative Models.** Tabular watermark involves two main functions. **(1) Generate**: Given a secret watermark key $k$, this function produces a watermarked table. Similar to standard generation, each row of this table is sampled $i.i.d.$, but from a distribution $p(\mathbf{x}, k)$. **(2) Detect**: Provided with a table and a specific key $k$, this function examines the table to determine if it carries the watermark associated with that particular key.

**Threat Model.** We consider the following watermarking protocol between three parties: the tabular data provider, the user, and the detector. **(1)** The tabular data provider shares a watermark key $k$ and certain metadata related to the data distribution (e.g., the maximum and minimum values of each column) with the detector. **(2)** The user asks the tabular data provider to generate a table $T$. **(3)** The user publishes a table $T'$, which can either be an (edited version of the) original table $T$ or an independent table. **(4)** The detector determines whether the table $T'$ is watermarked or not.)

## 3 METHOD

In this section, we introduce MUSE, a general paradigm for watermarking tabular data generators. We begin by outlining the paradigm's high-level architecture for generation and detection (Section 3.1). We then detail its core components: the scoring function (Section 3.2), which can be instantiated with different designs to balance trade-offs between detectability and distortion.

### 3.1 WATERMARK GENERATION AND DETECTION PARADIGM

We define the overall generation and detection process of our MUSE method in this section. The generation of each watermarked row can be decomposed into the following two steps:

**Generation.** The generation of each watermarked row is achieved through a two-phase process:

1. **Sample Candidates.** Generate a set of $m$ candidate rows by $i.i.d.$ sampling from the model's distribution $p(\mathbf{x})$.
2. **Select the Highest-Scoring Candidate.** Apply a *watermark scoring function* $s_k(\cdot)$ to each candidate $\mathbf{x}_i$ using watermark key $k$ and select the **highest-scoring candidate** (ties are broken randomly) as the watermarked row. We will detail the watermark scoring function in Section 3.2.

To produce a watermarked table with $N$ rows, we repeat the above process $N$ times. In practice, the selection procedure can be **fully parallelized** across the $N$ groups since each group contains $i.i.d.$ samples. The watermark generation process is illustrated in Figure 2 and Algorithm 1.

**Detection.** The generation process naturally creates a statistical artifact. By consistently selecting the highest-scoring sample, we ensure that a watermarked table will exhibit a significantly higher average score than an unwatermarked one. To detect the watermark, we formalize this intuition as follows: given a (watermarked or unwatermarked) table $T$ consists of $N$ rows: $T := (\mathbf{x}_1, \dots, \mathbf{x}_N)$, we compute the detection statistic:

$$S(T) = \frac{1}{N} \sum_{i=1}^{N} s_k(\mathbf{x}_i). \tag{1}$$

A table is flagged as watermarked if its mean score $S(T)$ surpasses a predefined threshold derived from the expected score of non-watermarked data. The formal statistical test is detailed in Appendix F.4.

## 3.2 Watermark Scoring Function

Our watermark scoring function, $s_k(\cdot)$, has two components: a score generation design, described in Section 3.2.1, and a column selection implementation, detailed in Section 3.2.2.

### 3.2.1 Score Generation Designs

Let $\pi(\mathbf{x})$ be a selection function that selects a subset of columns from a sample $\mathbf{x}$ (we will detail the design of the selection function in Section 3.2.2), with $\mathcal{J}$ being the set of selected column indices. We present two designs for generating a score from this selection and the watermark key $k$.

- **Joint-Vector (JV) Hashing**: Hashes the entire vector of selected values as a concatenated vector.

$$h = H\big(\pi(\mathbf{x}),, k\big), \quad s_k^{\mathrm{JV}}(\mathbf{x}) = f(h). \tag{2}$$

- **Per-Column (PC) Hashing**: Hashes each selected column value independently then aggregates.

$$h_i = H(\mathbf{x}_i, k) \;\; (i \in \mathcal{J}), \quad s_k^{\mathrm{PC}}(\mathbf{x}) = \frac{1}{|\mathcal{J}|} \sum_{i \in \mathcal{J}} f(h_i). \tag{3}$$

In both designs, $f$ is a pseudorandom function (PRF) whose output bit follows a Bernoulli(0.5) distribution. Intuitively, by placing equal probability mass on the two extreme values (0 and 1), this distribution provides maximal separation between binary signals (watermarked vs. non-watermarked). This intuition is rigorously established in Theorem 4.1.

**Robustness and Distortion Trade-off.** The choice between JV and PC hashing represents a fundamental trade-off between robustness against attacks and the preservation of the original data distribution (low distortion). The JV design excels at minimizing distortion. By hashing a concatenated vector of column values, it operates in a vast input space, making hash collisions rare and thus preserving the data's statistical properties. However, this "all-or-nothing" approach is fragile; a single modification to any of the selected columns can alter the entire hash, compromising the watermark signal for that sample. In contrast, the PC design prioritizes robustness. It embeds the watermark signal independently across multiple columns, ensuring that the overall signal can survive partial data deletion or modification. This resilience comes at the cost of a higher potential for distortion, as the smaller input space of individual columns can lead to more frequent hash collisions and a more concentrated statistical bias. We empirically validate this trade-off in our experiments (Section 5).

### 3.2.2 Column Selection Implementation

**Adaptive Selection for JV Hashing.** The selection strategy for Joint-Vector (JV) hashing must address two critical vulnerabilities. First, the design's "all-or-nothing" nature makes it fragile: any modification to a selected value invalidates the entire watermark, which necessitates the use of a sparse selection (a small number of columns) to minimize the attack surface. However, simply choosing a fixed sparse set of columns creates a predictable target for adversaries, who could nullify the watermark by altering just those few features. To overcome both challenges, we propose a strategy that fulfils both requirements. This is achieved by selecting columns based on their quantile rank, which measures a value's position relative to the empirical distribution of the training data. For each row $\mathbf{x}$ and each column index $j$, we compute its rank $r_j \in [0, 1]$:

$$r_j = \frac{v_j - v_{\min,j}}{v_{\max,j} - v_{\min,j}}, \tag{4}$$

where for a numerical column, $v_j$ equals the $j$-th column value of $\mathbf{x}$: $v_j := \mathbf{x}_j$ and $v_{\min,j}, v_{\max,j}$ are pre-computed min and max values from the training data. For a categorical column, $v_j$ is its ordinal index. Finally, for each sample $\mathbf{x}$, we take its per-column ranks $r_j$, sort them within the row, and select the columns whose positions match a fixed quantile set $\mathcal{Q}$.

**Full Selection for PC Hashing.** In contrast to the JV design, the Per-Column (PC) approach is inherently robust, as it aggregates watermark signals embedded independently across each column. This design ensures that modifications to a *subset* of columns do not corrupt the entire watermark. The overall signal's strength and resilience scale directly with the number of columns used. Therefore, to maximize robustness, the ideal strategy is to select *all available columns*. For this design, we configure $\pi(\mathbf{x}) := \mathbf{x}$ to simply use all features, setting the index set to $\mathcal{J} = \{1, \ldots, M\}$.

*Remark* 3.1 (Modification of All Columns). For datasets consisting of purely numerical columns, an adversary may inject small perturbations across *all* entries simultaneously. To mitigate the sensitivity of hashing to such noise, a preprocessing step $f$ (e.g., quantization) can be applied prior to hashing, ensuring that $f(x) \approx f(x + \epsilon)$ for a small noise term $\epsilon$. See Appendix E.2 for a complete discussion.

*Remark* 3.2 (Watermark Security). JV hashing selects a sparse subset of columns based on a fixed quantile set, which introduces a potential vulnerability: if the quantile set is leaked, an adversary can identify the watermark-carrying columns and scrub the watermark. In contrast, PV hashing is inherently more resilient since the watermark signal is spread across all columns. A simple way to improve the security of JV hashing is by applying a keyed pseudorandom permutation (PRP) $\pi_k$ to the column indices before quantile selection ($\mathbf{x} \mapsto \pi_k(\mathbf{x})$). Under this design, the watermark-carrying columns are indexed by the secret key $k$, and identifying them becomes computationally equivalent to breaking the underlying PRP. We further provide empirical evidence showing that recovering the quantile set is non-trivial in practice; See Appendix E.3 for a complete discussion.

## 4 ANALYSIS

In this section, we provide a theoretical analysis of the detectability and distribution-preserving properties of the MUSE paradigm.

### 4.1 CALIBRATING THE NUMBER OF REPEATED SAMPLES

Given the detection statistic Equation (1), we will show how the detectability of MUSE depends on (1) the number of watermarked samples $N$ and (2) the number of repeated samples $m$.

**Theorem 4.1** (Watermark Calibration Guarantees). *Denote a watermarked table as $T_{\text{wm}}$ and an unwatermarked table as $T_{\text{no-wm}}$, each consisting of $N$ rows. Let $\mathbf{x} \sim p(\mathbf{x})$ be a random variable drawn from the data distribution, and let $\mathbf{x}_1, \dots, \mathbf{x}_m$ be i.i.d. samples from $p(\mathbf{x})$. Define $\mu_{\text{no-wm}} = \mathbb{E}_{\mathbf{x} \sim p(\mathbf{x})}[s_k(\mathbf{x})]$ as the expected score of an unwatermarked sample, and define $\mu_{\text{wm}}^m = \mathbb{E}_{\mathbf{x}_i \sim p(\mathbf{x})} \left[ \max_{i \in [m]} s_k(\mathbf{x}_i) \right]$ as the expected score of a watermarked sample obtained via $m$ repeated samples. Suppose the scoring function satisfies $s_k(\cdot) \in [0, 1]$, we have:*

1. *The False Positive Rate (FPR) of the watermark detection is upper bounded:*

$$\Pr\left( S(T_{\text{no-wm}}) > S(T_{\text{wm}}) \right) \leq \exp\left( -\frac{N \cdot (\mu_{\text{wm}}^m - \mu_{\text{no-wm}})^2}{2} \right). \tag{5}$$

2. *The RHS of the bound is minimized when $s_k(\mathbf{x})$ follows a $\text{Bernoulli}(0.5)$ distribution.*

3. *Under this optimal distribution, let $N > 8\log(1/\alpha)$, then to ensure the FPR does not exceed a target threshold $\alpha$, it suffices to set the number of repeated samples $m$ as:*

$$m = \max\left( 2, \left\lceil \log_{0.5}\left( 0.5 - \sqrt{\frac{2\log(1/\alpha)}{N}} \right) \right\rceil \right). \tag{6}$$

Theorem 4.1 enables MUSE to calibrate the number of repeated samples $m$ to achieve a target false positive rate with theoretical guarantees. This allows the method to embed *just enough* watermarking signal to ensure the desired detectability. Intuitively, since no redundant watermarking signal is embedded, the impact of watermarking on the generation quality is minimal. In Figure 3, we plot $m$ as a function of table size $N$ for various target FPRs, based on Equation (6) (omitting the ceiling operation for clarity). We observe that $m$ quickly saturates as $N$ increases. For instance, to achieve a 0.01% FPR, $m = 2$ suffices when $N \geq 300$, and even for $N = 100$, $m = 4$ is enough. In the rest of the paper, MUSE's $m$ is set by Equation (6) unless otherwise specified.

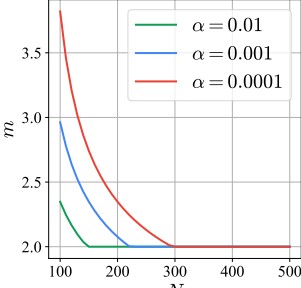

Figure 3: $m$ vs. $N$ under different $\alpha$ values (smoothed).

### 4.2 DISTRIBUTION-PRESERVATION

An effective watermarking algorithm must not compromise the quality of the generated data, a requirement formalized in domains like image (Gunn et al., 2024) and text generation (Kuditipudi et al.,

---

**Algorithm 1** MUSE Watermark Generation

---

1: **Input:** watermark key $k$, a generative model $p(\mathbf{x})$, False Positive Rate $\alpha$, number of target watermarked samples $N$
2: **Output:** watermarked table $T_{wm}$
3: Compute the number of repeated samples $m$ based on $N$ and $\alpha$ via Equation (6) ▷ Calibration.
4: Get $m \cdot N$ i.i.d. samples from $p(\mathbf{x})$ and divide them into $N$ groups: $(\mathcal{G}_i)_{i=1}^N$.
5: Initialize a list $T_{wm}$ to store the watermarked table
6: **for** $i \leftarrow 1$ to $N$ **do**            ▷ Fully parallelizable.
7:    $\mathbf{x}_1, \ldots, \mathbf{x}_m \leftarrow \mathcal{G}_i$
8:    **for** $t \in \{1, \ldots, m\}$ **do**
9:       Select columns for $\mathbf{x}_t$ with strategy in Section 3.2.2      ▷ Column selection.
10:       Compute the score for $\mathbf{x}_t$ with strategy in Section 3.2.1 to get $s_t$    ▷ Score generation.
11:    **end for**
12:    $i \leftarrow \arg\max_{t \in \{1, \ldots, m\}} s_t$        ▷ Selection of the highest-scoring sample.
13:    Append $\mathbf{x}_i$ to $T_{wm}$
14: **end for**
15: **return** $T_{wm}$

---

2023). For tabular data generation, we adapt this requirement by demanding that the watermarking process preserves the original data distribution, which we formalize as follows:

**Definition 4.2** (Multi-Sample Distribution-Preservation). Denote the space of watermark keys as $\mathcal{K}$ and the original data distribution as $p_{\text{data}}(\mathbf{x})$. Let $(\tilde{\mathbf{x}}_1, \ldots, \tilde{\mathbf{x}}_N)$ be a sequence of $N$ samples generated consecutively by a watermarking algorithm $\Gamma$ using the same key $k \sim \text{Unif}(\mathcal{K})$. The algorithm $\Gamma$ is *multi-sample distribution-preserving* if for any $N > 0$, it satisfies:

$$\mathbb{P}_{k \sim \text{Unif}(\mathcal{K})}(\tilde{\mathbf{x}}_1, \ldots, \tilde{\mathbf{x}}_N) = \prod_{i=1}^N p_{\text{data}}(\tilde{\mathbf{x}}_i). \tag{7}$$

Our algorithm attains the multi-sample distribution-preserving property through a mechanism we call *Repeated Column Masking*. The key idea is to cache the history of column values that have previously been selected for watermark embedding. When processing a new sample, if its candidate column value has already been used for watermarking, the algorithm skips embedding on that sample. This safeguard prevents systematic bias from repeated column reuse across samples. The design is inspired by the *repeated key masking* technique in LLM watermarking, which ensures sequence-level distribution-preserving guarantees (Hu et al., 2023; Dathathri et al., 2024). Formally, we have:

**Theorem 4.3.** *Let $m = 2$. The watermarking process in Algorithm 1, augmented with repeated column masking, satisfies multi-sample distribution-preserving as defined in Definition 4.2.*

*Remark* 4.4. While the repeated column masking mechanism ensures distribution-preserving, it introduces a practical trade-off. By design, this mechanism chooses to skip the watermarking process when repeated column values are detected, which in turn weakens the watermark's detectability. We empirically validate this trade-off in our ablation studies (Section 5.4).

## 5 EXPERIMENTS

In this section, we provide a comprehensive empirical evaluation of MUSE. We aim to answer the following research questions. **Q1: Detectability v.s. Distribution Preservation** (Section 5.2): Can MUSE achieve strong detectability while preserving the distribution of the generated data? **Q2: Robustness** (Section 5.3): How resilient is the watermark to a range of post-processing attacks, such as row/column deletion or value perturbation? **Q3: Component-wise Analysis** (Section 5.4): How does MUSE perform under different design choices of its components?

### 5.1 SETUP

**Datasets.** We consider four real-world tabular datasets containing both numerical and categorical attributes: `Adult`, `Default`, `Shoppers`, and `Beijing` and two datasets with only numerical attributes: `California` and `Letter`. Detailed dataset statistics are provided in Appendix F.2.

**Evaluation Protocols. (1) Detectability**: To evaluate the detectability of the watermark, we report the area under the curve (AUC) of the receiver operating characteristic (ROC) curve, and the True

Table 1: **Watermark generation quality and detectability**, ▨ indicates best performance, ▨ indicates second-best performance. ↑ indicates higher is better, ↓ indicates lower is better. The performance gain is computed with respect to the best performing baseline.

| Dataset | Method | Watermark Generation Quality | | | | Watermark Detectability | | | |
|---|---|---|---|---|---|---|---|---|---|
| | | Num. Training Rows | | | | 100 | | 500 | |
| | | **Marg.** (↑) | **Corr.** (↑) | **C2ST** (↑) | **MLE Gap** (↓) | AUC | T@0.1%F | AUC | T@0.1%F |
| Adult | w/o WM | 0.994 | 0.984 | 0.996 | 0.017 | - | - | - | - |
| | TR | 0.919 | 0.870 | 0.676 | 0.046 | 0.590 | 0.004 | 0.774 | 0.171 |
| | GS | 0.751 | 0.619 | 0.058 | 0.084 | 1.000 | 1.000 | 1.000 | 1.000 |
| | TabWak | 0.935 | 0.885 | 0.769 | 0.048 | 0.844 | 0.089 | 0.990 | 0.592 |
| | TabWak* | 0.933 | 0.879 | 0.713 | 0.085 | 0.999 | 0.942 | 1.000 | 1.000 |
| | **MUSE-JV** | 0.979 (+74.6%) | 0.963 (+78.8%) | 0.883 (+50.2%) | 0.017 (+63.0%) | 1.000 | 1.000 | 1.000 | 1.000 |
| | **MUSE-PC** | 0.953 (+30.5%) | 0.925 (+40.4%) | 0.790 (+9.3%) | 0.018 (+60.9%) | 1.000 | 1.000 | 1.000 | 1.000 |
| Default | w/o WM | 0.990 | 0.934 | 0.979 | 0.000 | - | - | - | - |
| | TR | 0.895 | 0.888 | 0.564 | 0.161 | 0.579 | 0.001 | 0.848 | 0.034 |
| | GS | 0.701 | 0.678 | 0.059 | 0.182 | 1.000 | 1.000 | 1.000 | 1.000 |
| | TabWak | 0.911 | 0.902 | 0.568 | 0.156 | 0.896 | 0.071 | 0.997 | 0.611 |
| | TabWak* | 0.906 | 0.894 | 0.550 | 0.176 | 0.965 | 0.218 | 1.000 | 0.995 |
| | **MUSE-JV** | 0.983 (+91.1%) | 0.925 (+71.9%) | 0.963 (+96.1%) | 0.002 (+98.7%) | 1.000 | 1.000 | 1.000 | 1.000 |
| | **MUSE-PC** | 0.960 (+62.0%) | 0.920 (+56.3%) | 0.866 (+72.5%) | 0.003 (+98.1%) | 1.000 | 1.000 | 1.000 | 1.000 |
| Shoppers | w/o WM | 0.985 | 0.974 | 0.974 | 0.017 | - | - | - | - |
| | TR | 0.888 | 0.880 | 0.501 | 0.077 | 0.575 | 0.001 | 0.830 | 0.058 |
| | GS | 0.729 | 0.688 | 0.061 | 0.154 | 1.000 | 1.000 | 1.000 | 1.000 |
| | TabWak | 0.903 | 0.886 | 0.548 | 0.132 | 0.860 | 0.106 | 0.990 | 0.353 |
| | TabWak* | 0.897 | 0.879 | 0.525 | 0.384 | 0.742 | 0.002 | 0.981 | 0.185 |
| | **MUSE-JV** | 0.982 (+96.3%) | 0.974 (+100.0%) | 0.950 (+94.4%) | 0.015 (+80.5%) | 1.000 | 1.000 | 1.000 | 1.000 |
| | **MUSE-PC** | 0.962 (+72.0%) | 0.947 (+69.3%) | 0.871 (+75.8%) | 0.025 (+67.5%) | 1.000 | 1.000 | 1.000 | 1.000 |
| Beijing | w/o WM | 0.977 | 0.958 | 0.934 | 0.199 | - | - | - | - |
| | TR | 0.914 | 0.873 | 0.734 | 0.396 | 0.577 | 0.000 | 0.548 | 0.007 |
| | GS | 0.656 | 0.529 | 0.097 | 0.715 | 1.000 | 1.000 | 1.000 | 1.000 |
| | TabWak | 0.923 | 0.871 | 0.792 | 0.375 | 0.925 | 0.096 | 0.999 | 0.978 |
| | TabWak* | 0.917 | 0.860 | 0.761 | 0.403 | 0.996 | 0.734 | 1.000 | 1.000 |
| | **MUSE-JV** | 0.972 (+90.7%) | 0.955 (+96.5%) | 0.926 (+94.4%) | 0.209 (+44.3%) | 1.000 | 1.000 | 1.000 | 1.000 |
| | **MUSE-PC** | 0.963 (+74.1%) | 0.943 (+82.4%) | 0.898 (+74.6%) | 0.213 (+43.2%) | 1.000 | 1.000 | 1.000 | 1.000 |

Figure 4: The tradeoff between average $z$-statistic and data fidelity (computed as average of Marg., Corr., C2ST and MLE) under different number of repeated sample $m$.

Positive Rate when the False Positive Rate is at 0.1%, denoted as *TPR@0.1%FPR*. **(2) Distribution Preservation**: To evaluate the distribution-preserving ability of the watermarked data, we follow standard fidelity and utility metrics used in tabular data generation (Zhang et al., 2024c; Kotelnikov et al., 2023): we report Marginal distribution (Marg.), Pair-wise column correlation (Corr.), Classifier-Two-Sample-Test (C2ST), and Machine Learning Efficiency (MLE). For MLE, we report the gap between the downstream task performance of the generated data and the real test set (MLE Gap). We refer the readers to Section F.3 for a more detailed definition of each evaluation metric. **(3) Robustness**: We evaluate the robustness of the watermarked data against five representative post-processing attacks. In addition, we also consider an *adaptive adversary* who tries to reverse-engineer the watermark scheme. Detailed description will be presented in Section 5.3.

**Baselines and Implementation Details.** We compare our method with TabWak (Zhu et al., 2025) and its improved variant TabWak*, the only existing generative watermarking approach for tabular data, using their official implementations. We also include two image watermarking methods, TreeRing (Wen et al., 2023) and Gaussian Shading (Yang et al., 2024), as auxiliary baselines (see Appendix F.5 for detailed implementation). For completeness, we also include two edit-based methods: TabularMark (Zheng et al., 2024) and WGTD (He et al., 2024), with detailed results in Section C.2. All experiments use TabSyn (Zhang et al., 2024c) as the tabular generative model trained with the official codebase. Notably, the official TabWak implementation bypasses *quantile*

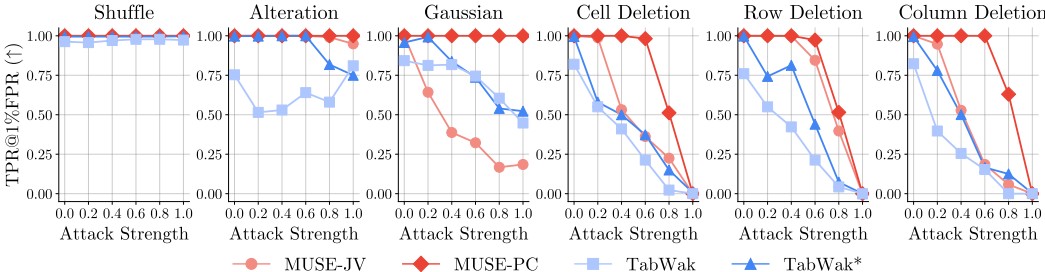

Figure 5: Detection performance of watermarking methods against different types of tabular data attacks across varying attack intensities. The results are averaged over all datasets.

*normalization inversion*, assuming access to ground-truth data unavailable at detection time, which may favor its performance (see Section D.2 for more discussion). Generation quality is evaluated across ten repetitions, and we report the averaged results.

## 5.2 DETECTABILITY AND DISTRIBUTION PRESERVATION

We address the first question: whether the watermarking method achieves high watermark detectability while introducing minimal distortion to the generated data. Based on experiments results in Table 1, our obervations are summarized as follows:

**(1)** Regarding generation quality, both MUSE variants consistently outperform the baselines across all datasets. The MUSE-JV variant is particularly effective, reducing distortion rates on fidelity metrics (Marg., Corr., C2ST) by $84 - 88\%$ compared to the best performing baselines. In contrast, all inversion-based methods suffer from significant data distortion. We attribute this to the error accumulation inherent in their recovery process: to ensure a watermark can be detected after a noisy inversion, the initial signal must be excessively strong, which inherently leads to large distortion. **(2)** In terms of detectability, both variants of MUSE achieve perfect detection performance across all datasets, as measured by both AUC and T@0.1%F. While GS also achieves strong detection scores, this comes at the cost of significantly higher distortion across all fidelity metrics. **(3)** The JV variant achieves better fidelity metrics than the PC variant. We will show in the next section that the PC variant is more robust to post-processing attacks. In Figure 4, we visualize the tradeoff between detectability ($z$-stat) and data fidelity (computed as the average of Marg., Corr., C2ST, and MLE). Consistent with the theoretical analysis in Theorem 4.1, increasing $m$ in both MUSE variants leads to stronger detectability but degrades data fidelity. The results also demonstrate that, for a fixed $m$, PC hashing generally yields higher detectability than JV-hashing (with the exception of the `Shoppers` dataset at $m \in \{2, 4\}$), albeit at the cost of lower data fidelity. This empirically validates the design principles behind these two hashing strategies.

## 5.3 ROBUSTNESS AGAINST ATTACKS

**Post-processing Attacks.** We evaluate robustness against six common transformations in tabular data: *row shuffling*, *row deletion*, *column deletion*, *cell deletion*, *value alteration*, and *Gaussian perturbation*. Attacks are applied at perturbation levels from 0.0 to 1.0 in 0.2 increments. Deletion-based attacks replace a fraction of rows, columns, or cells with unwatermarked samples from the same generative model. Value alteration perturbs numerical entries by multiplying them with scalars from $(0.8, 1.2)$, while row shuffling permutes a subset of rows. For the Gaussian perturbation attack, each numeric value is perturbed by zero-mean noise whose standard deviation is the perturbation level times the magnitude of that value. We benchmark the detectability of MUSE-JV and MUSE-PC against TabWak and TabWak* on all mixed-type datasets ($N{=}500$, $m{=}2$), and additionally assess the robustness of MUSE-PC under Gaussian perturbation on the two numerical-only datasets. As shown in Figure 5 and Appendix E.2, MUSE-JV matches or surpasses TabWak and TabWak* in five of six post-processing attacks, while the **PC variant achieves the strongest robustness across all settings**. The superior resilience of the PC design, contrasted with the higher fidelity of the JV design, illustrates the fundamental trade-off between robustness and distortion. **The capability to select the desired point on the tradeoff spectrum underscores the inherent flexibility of our framework.**

**Adaptive Attacks.** We assess the robustness of MUSE against adaptive adversaries attempting to reverse-engineer the watermark. Specifically, we focus on *spoofing attacks* (Sadasivan et al., 2023),

Table 3: **Component-wise ablation study** of MUSE. All experiments are conducted on the `Adult` dataset (with 15 columns). For detectability, we report the $z$-statistic (defined in Section F.4). Each color block indicates a different component of the method. ↑ indicates higher is better.

| Hashing | Model | PRF. | Mask | Num. Col. | $z$-stat.↑ | Marg.↑ | Corr.↑ | C2ST↑ |
|---------|-------|------|------|-----------|-----------|--------|--------|-------|
| JV | TabSyn | Bernoulli | No | 3 | 7.348 | **0.979** | **0.963** | **0.883** |
| JV | TabDAR | Bernoulli | No | 3 | 7.270 | 0.977 | 0.958 | 0.880 |
| JV | DP-TBART | Bernoulli | No | 3 | **7.544** | 0.951 | 0.931 | 0.759 |
| JV | TabSyn | Bernoulli | No | 3 | **7.348** | **0.979** | **0.963** | **0.883** |
| JV | TabSyn | Uniform | No | 3 | 5.012 | 0.964 | 0.940 | 0.808 |
| PC | TabSyn | Bernoulli | No | 15 | **20.001** | **0.953** | **0.925** | **0.790** |
| PC | TabSyn | Uniform | No | 15 | -11.164 | 0.937 | 0.912 | 0.788 |
| JV | TabSyn | Bernoulli | No | 3 | **7.348** | 0.979 | 0.963 | 0.883 |
| JV | TabSyn | Bernoulli | Yes | 3 | 4.819 | **0.985** | **0.973** | **0.940** |
| PC | TabSyn | Bernoulli | No | 3 | 16.505 | **0.958** | **0.937** | **0.826** |
| PC | TabSyn | Bernoulli | Yes | 7 | 19.998 | 0.950 | 0.929 | 0.797 |
| PC | TabSyn | Bernoulli | No | 15 | **20.001** | 0.953 | 0.925 | 0.790 |

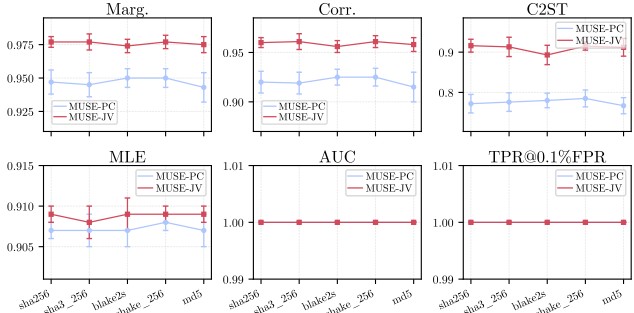

Figure 6: Sensitivity results for different hash families. Error bars denote standard deviation over the key space. MUSE remains insensitive to both hash functions and key space.

Figure 7: Impact of the number of watermarked rows $N$ on detectability, which increases monotonically as more rows are watermarked.

where the attacker's goal is to produce samples that can be falsely claimed as watermarked, without knowing the secret keys of the watermark. Instead of developing bespoke, scheme-specific spoofing attacks (Jovanović et al., 2024), we adopt a general distillation-based spoofing framework (Sander et al., 2024; Gu et al., 2023): an adversary trains a strong generative model

Table 2: Adaptive attack results.

| | 100 Rows | | 500 Rows | |
|---------|----------|---------|----------|---------|
| Dataset | AUC | T@0.1%F | AUC | T@0.1%F |
| Adult | 0.465 | 0.01 | 0.566 | 0.02 |
| Default | 0.599 | 0.01 | 0.708 | 0.02 |
| Shoppers | 0.683 | 0.03 | 0.866 | 0.41 |
| Beijing | 0.470 | 0.00 | 0.581 | 0.05 |

(e.g., TabSyn) directly on the watermarked data, attempting to absorb and reproduce its statistical structure. The spoofing attack is successful if the generated samples from the trained model are detected as watermarked. The results in Table 2 demonstrate that the adversarial model largely fails to replicate the watermark. On three of the four datasets (`Adult`, `Default`, and `Beijing`), its generated output is statistically indistinguishable from clean data (AUC ≈ 0.5 and T@0.1%F ≈ 0.00). While a faint signal is detected on the `Shoppers` dataset, the watermark is severely degraded. This failure of a powerful generative model to passively learn the watermark's patterns provides strong evidence for MUSE's resilience against reverse-engineering attacks.

## 5.4 ABLATION STUDY AND FURTHER ANALYSIS

We perform a component-wise ablation to evaluate the contribution of each design choice in our watermarking framework. All experiments are conducted on the `Adult` dataset, and we generate watermarked tables with $N = 100$ rows unless otherwise noted. For detectability, we report the

$z$-statistic, which quantifies how many standard deviations the observed detection score deviates from its null expectation (no watermark). The exact formulas for JV and PC are given in Section F.4.

**Impact of Score Function.** We compare two scoring distributions: (1) a Bernoulli distribution with mean $0.5$, and (2) a uniform distribution over $[0, 1]$. For both the JV and PC hashing designs, the Bernoulli score consistently achieves superior detectability, as shown in Table 3. This result is consistent with our theoretical analysis in Lemma G.2, which identifies Bernoulli$(0.5)$ as the optimal scoring distribution for our detection formulation.

**Impact of the Number of Selected Columns.** For the PC design, the number of selected columns presents a trade-off between detectability and data quality. As shown in Table 3, using more columns boosts detectability by strengthening the aggregated watermark signal. However, this also raises the potential for distortion, as more frequent hash collisions on small column value spaces can introduce a concentrated statistical bias.

**Impact of Repeated Column Masking.** The repeated column masking mechanism is designed to enforce the formal distribution-preserving property of our watermark, thereby maintaining high data quality. To quantify its impact, we ablate this component for both our JV and PC designs. As shown in Table 3, enabling masking improves data fidelity at the cost of a reduction in detectability.

**Model-Agnostic Applicability.** While our main experiments use a diffusion model (Zhang et al., 2024c), MUSE is a model-agnostic framework. To validate this, we apply it to two other diverse generative paradigms: an autoregressive model (DP-TBART (Castellon et al., 2023)) and a masked generative model (TabDAR (Zhang et al., 2024a)). As shown in Table 3, MUSE consistently achieves high detectability and data fidelity across all three model families, confirming its broad applicability.

**Computation Time.** We compare the effective watermarking time (generation + detection) of MUSE with baselines that rely on DDIM inversion. We generate 10K watermarked rows of the `Adult` dataset. As shown in Figure 8, MUSE achieves significantly lower detection time by avoiding the costly inversion process. Notably, its generation time is also lower than that of the baselines, despite using multi-sample generation ($m = 2$). This efficiency arises from MUSE's compatibility with fast score-based diffusion mod-

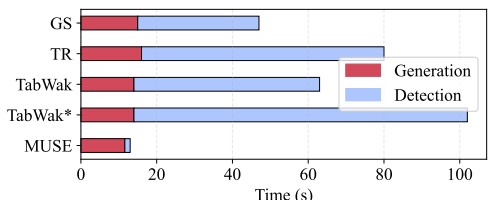

Figure 8: Watermark generation and detection time of MUSE and inversion-based baselines.

els (Zhang et al., 2024c; Karras et al., 2022), which require only 50 sampling steps. Conversely, the inversion-based baselines must use a much slower 1,000-step process for both generation and detection (Zhu et al., 2025).

**Sensitivity analysis on hash function and key.** We evaluate MUSE on the `Adult` dataset ($N{=}500, m{=}2$) across five hash families (SHA-256, SHA3-256, BLAKE2s, SHAKE-256, MD5) and 13 randomly sampled keys with bit-lengths from 32 to 128 bits, measuring detectability (AUC, TPR@0.1%FPR) and fidelity (Marg., Corr., C2ST, MLE). As shown in Figure 6, detectability remains perfect (AUC=1.0) across all configurations and fidelity variance is negligible (std $\approx 0.01$), confirming that MUSE is robust to the choice of hash function and secret key.

## 6 CONCLUSION

We propose MUSE, a model-agnostic watermarking method that embeds signals via multi-sample selection, eliminating the need for inversion. MUSE achieves strong detectability with minimal distribution shift and scales across diverse generative models. Extensive experiments demonstrate its superiority over existing methods in both generation quality and watermark detectability. As synthetic tabular data becomes increasingly adopted in high-stakes domains, MUSE offers a practical and generalizable safeguard for data provenance, ownership verification, and misuse detection.

## 7 ACKNOWLEDGMENT

This work is supported in part by NSF under grants III-2106758, and POSE-2346158.

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

# APPENDIX

## A  THE USE OF LARGE LANGUAGE MODELS (LLMS)

In the preparation of this manuscript, we utilized Large Language Model (LLM) as a general-purpose assistive tool. The primary applications of the LLM were for polishing the writing, including improving grammar, clarity, and conciseness of the text. Additionally, the LLM was used to generate boilerplate code for setting up and running experiments, which helped accelerate the implementation process.

The LLM did not contribute to the core research ideation, the development of the proposed methodology, the analysis of the results, or the scientific conclusions presented in this paper. All content, including the final text and experimental code, was reviewed, edited, and validated by the authors, who take full responsibility for the accuracy and integrity of this work.

## B  RELATED WORK

**Generative Watermarking.** Generative watermarking embeds watermark signals during the generation process, typically by manipulating the generation randomness through pseudorandom seeds. This approach has proven effective and efficient for watermarking in image, video, and large language model (LLM) generation. In image and video generation, where diffusion-based models are the *de facto* standard, watermarking methods inject structured signals into the noise vector in latent space (Wen et al., 2023; Yang et al., 2024; Huang et al., 2024). Detection involves inverting the diffusion sampling process (Dhariwal and Nichol, 2021; Hong et al., 2024; Pan et al., 2023) to recover the original noise vector and verify the presence of the embedded watermark. For LLMs, generative watermarking methods fall into two categories: (1) *Watermarking during logits generation*, which embeds signals by manipulating the model's output logits distribution (Kirchenbauer et al., 2023a; Zhao et al., 2023a; Hu et al., 2023; Dathathri et al., 2024; Giboulot and Furon, 2024; Liu et al., 2023; 2024a); and (2) *Watermarking during token sampling*, which preserves the logits distribution but replaces the stochastic token sampling process (e.g., multinomial sampling) with a pseudorandom procedure seeded for watermarking (Aaronson and Kirchner, 2022; Kuditipudi et al., 2023; Christ et al., 2024). In this sense, sampling-based watermarking is conceptually similar to inversion-based watermarking used in diffusion models. We refer the reader to (Liu et al., 2024b; Pan et al., 2024) for a comprehensive survey of watermarking for LLMs. Bahri and Wieting (2024), SynthID (Dathathri et al., 2024), and WaterMax (Giboulot and Furon, 2024) similarly explore watermarking via repeated candidate sampling. However, the distinct nature of tabular data necessitates a fundamentally different technical approach compared to text. First, the generative structures differ: text watermarking operates on a conditional 1D distribution (next-token-prediction), relying on a prefix window of context for hashing (Kirchenbauer et al., 2023b). In contrast, tabular models generate full rows i.i.d. from a multi-dimensional unconditional distribution (Kotelnikov et al., 2023; Zhang et al., 2024c;a), lacking the sequential history required for prefix-based hashing. While context-independent methods like Unigram (Zhao et al., 2023b) eliminate prefix reliance, applying their fixed Green-Red vocabulary split to tables introduces severe distributional distortion by permanently banning a subset of values across all columns. Second, the threat models diverge significantly: while text methods target token-level edits (insertion, substitution), tabular watermarking must withstand attacks unique to its data structure, such as row/column shuffling, row/column/cell deletion, and numerical value perturbation.

**Watermarking for Tabular Data** Traditional tabular watermarking techniques are edit-based, injecting signals by modifying existing data values. WGTD (He et al., 2024) embeds watermarks by altering the fractional parts of continuous values using a green list of intervals, but it is inapplicable to categorical-only data. TabularMark (Zheng et al., 2024) perturbs values in a selected numerical column using pseudorandom domain partitioning, but relies on access to the original table for detection, limiting its robustness in adversarial settings. Another significant drawback of such methods is the potential to distort the original data distribution or violate inherent constraints. To overcome this, TabWak (Zhu et al., 2025) introduced the first generative watermarking approach for tabular data. Analogous to inversion-based watermarks in diffusion models, TabWak embeds detectable patterns into the noise vector within the latent space. It also employs a self-clone and shuffling technique to minimize distortion to the data distribution. While TabWak avoids post-hoc editing, its reliance on inverting both the sampling process (e.g., DDIM (Song et al., 2020b)) and preprocessing steps (e.g.,

Table 4: Watermark generation quality and detectability, ▇ indicates best performance, ▇ indicates second-best performance. For clarity, only our method is highlighted in detection.

| Dataset | Method | Watermark Generation Quality | | | | Watermark Detectability | | | |
|---|---|---|---|---|---|---|---|---|---|
| | | Num. Training Rows | | | | 100 | | 500 | |
| | | Marg.z↑ | Corr.↑ | C2ST↑ | MLE Gap↓ | AUC | T@0.1%F | AUC | T@0.1%F |
| Adult | w/o WM | 0.994 | 0.984 | 0.996 | 0.017 | - | - | - | - |
| | TabularMark | 0.983 | 0.949 | 0.987 | 0.021 | 1.000 | 1.000 | 1.000 | 1.000 |
| | WGTD | 0.987 | 0.972 | 0.978 | 0.019 | 1.000 | 1.000 | 1.000 | 1.000 |
| | **MUSE-JV** | 0.979 | 0.963 | 0.883 | 0.017 | 1.000 | 1.000 | 1.000 | 1.000 |
| Beijing | w/o WM | 0.977 | 0.958 | 0.934 | 0.199 | - | - | - | - |
| | TabularMark | 0.935 | 0.789 | 0.941 | **0.528** | 1.000 | 1.000 | 1.000 | 1.000 |
| | WGTD | 0.964 | 0.948 | 0.929 | 0.527 | 1.000 | 1.000 | 1.000 | 1.000 |
| | **MUSE-JV** | 0.972 | 0.955 | 0.926 | 0.209 | 1.000 | 1.000 | 1.000 | 1.000 |
| Default | w/o WM | 0.990 | 0.934 | 0.979 | 0.000 | - | - | - | - |
| | TabularMark | 0.987 | 0.939 | 0.961 | 0.004 | 1.000 | 1.000 | 1.000 | 1.000 |
| | WGTD | 0.989 | 0.913 | 0.919 | 0.000 | 1.000 | 1.000 | 1.000 | 1.000 |
| | **MUSE-JV** | 0.983 | 0.925 | 0.963 | 0.002 | 1.000 | 1.000 | 1.000 | 1.000 |
| Shoppers | w/o WM | 0.985 | 0.974 | 0.974 | 0.017 | - | - | - | - |
| | TabularMark | 0.974 | 0.930 | 0.975 | 0.013 | 1.000 | 1.000 | 1.000 | 1.000 |
| | WGTD | 0.964 | 0.944 | 0.887 | 0.008 | 1.000 | 1.000 | 1.000 | 1.000 |
| | **MUSE-JV** | 0.982 | 0.974 | 0.950 | 0.015 | 1.000 | 1.000 | 1.000 | 1.000 |

quantile normalization (Wikipedia contributors, 2025)) can introduce reconstruction errors. These errors will in turn impair the watermark's detectability.

## C  ADDITIONAL EXPERIMENTS RESULTS

### C.1  OMITTED RESULTS ON ROBUSTNESS

We present the omitted robustness results in Figure 9, where MUSE is compared against TabWak and TabWak* on the `Adult`, `Beijing`, `Default`, and `Shoppers` datasets. Overall, MUSE demonstrates stronger robustness under cell deletion and row deletion attacks, while achieving comparable performance on alteration and column deletion attacks. Both MUSE and TabWak/TabWak* remain resilient to shuffle attacks, due to embedding watermarks at the individual row level. Notably, we observe that TabWak and TabWak* exhibit instability on certain datasets, such as `Shoppers` and `Beijing`, where detection performance fluctuates—first decreasing and then increasing—as attack intensity increases. We hypothesize that this behavior stems from the inherent instability of the VAE inversion process.

### C.2  OMITTED RESULTS ON EDIT-BASED WATERMARKING

We compare our method against two representative **edit-based** watermarking baselines, which embed watermarks by directly altering table entries. Since the official implementations of these methods are not publicly available, we reimplement them based on the descriptions in their original papers. We first outline their core methodologies and our reimplementation details, then present the comparative results in Table 4. **Our reproduced codes are provided in the supplementary material.** Below are the detailed implementations of the baselines.

**WGTD** (He et al., 2024). WGTD embeds watermarks by modifying the fractional part of continuous data points, replacing them with values from a predefined green list. Consequently, **it is limited to continuous data and cannot be applied to tables containing only categorical features**.

The watermarking process in WGTD involves three main steps: (i) dividing the interval $[0, 1]$ into $2m$ equal sub-intervals to form $m$ pairs of consecutive intervals; (ii) randomly selecting one interval from each pair to construct a set of $m$ "green list" intervals; and (iii) replacing the fractional part of each data point with a value sampled from the nearest green list interval, if the original does not already fall within one. Detection is performed via a hypothesis-testing framework that exploits the statistical properties of the modified distribution to reliably identify the presence of a watermark. For reproducibility, we adopt the original hyperparameter setting with $m = 5$ green list intervals.

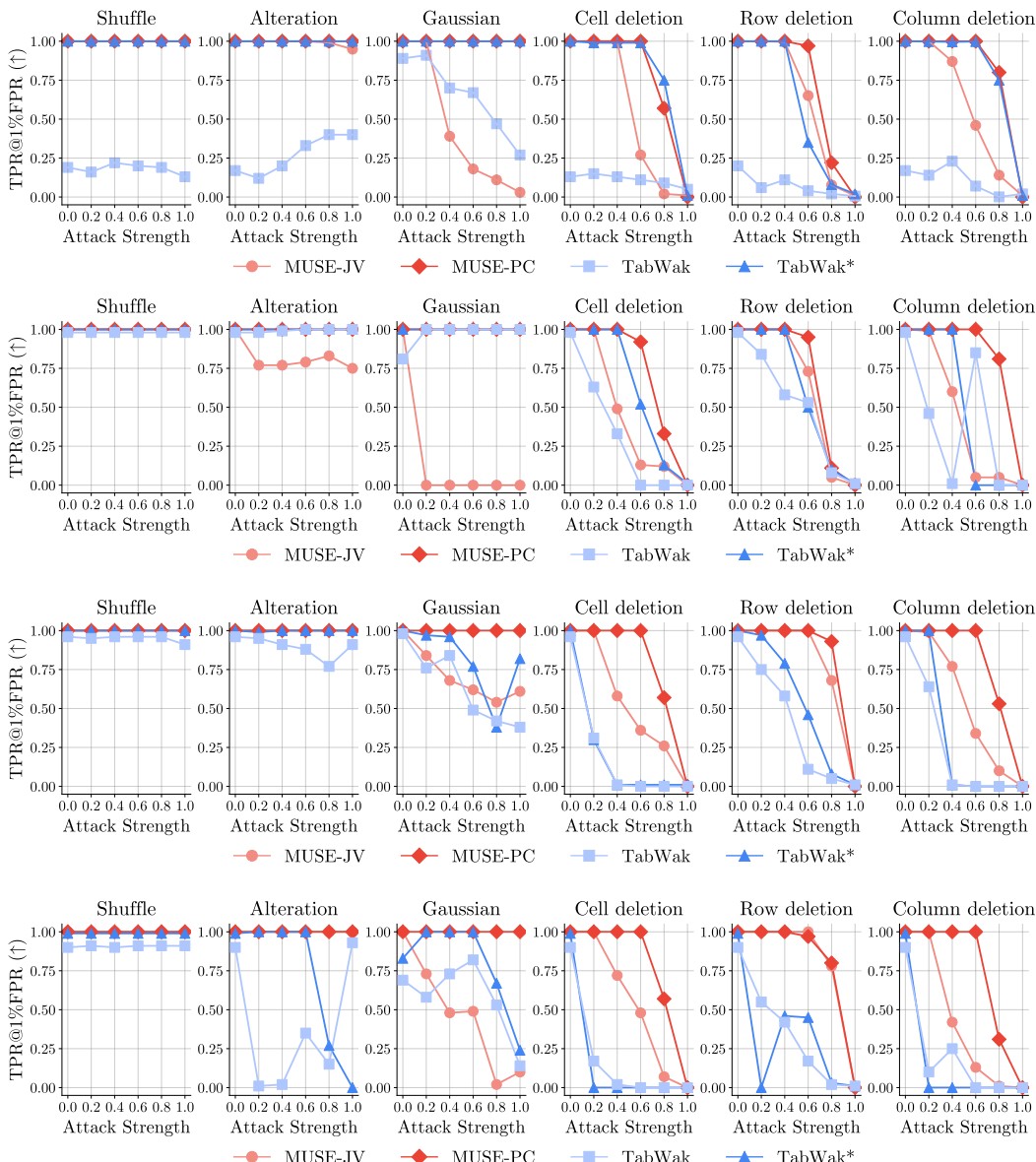

Figure 9: Detection performance of MUSE vs. TabWak/TabWak* against different types of tabular data attacks across varying attack intensities. From top to bottom: `Adult, Beijing, Default` and `Shoppers`.

**TabularMark** (Zheng et al., 2024). TabularMark embeds watermarks by perturbing specific cells in the data. It first pick a selected attribute/column to embed the watermark, then it generate pesudorandom partition of a fixed range into multiple unit domains, and label them with red and green domains, and finally perturb the selected column with a random number from the green domain. In our implementation, we choose the first numerical column as the selected attribute, and set the number of unit domains $k = 500$, the perturbation range controlled by $p = 25$, and configure $n_w$ as 10% of the total number of rows.

During detection, TabularMark leverages the original unwatermarked table to reverse the perturbations and verify whether the restored differences fall within the green domain. However, **this approach assumes access to the original unwatermarked table**, which is often impractical, especially in scenarios where the watermarked table can be modified by adversaries.

**Discussions.** As demonstrated in Table 4, both WGTD and TabularMark exhibit strong detection performance across all datasets. Furthermore, their generation quality is generally comparable to that of MUSE. However, a notable observation is the significant performance degradation measured by the MLE metric for both WGTD and TabularMark on the Beijing dataset (highlighted in bold). We hypothesize that this performance drop stems from the post-editing process, which may introduce substantial artifacts into the data. These artifacts, in turn, could negatively impact the performance of downstream machine learning tasks.

### C.3 Visualization of statistical signal

Intuitively, our method embeds watermarks by biasing the score distribution towards high score values. In this section, we provide visualizations that directly illustrate the statistical signal introduced by our watermark in both the JV and PC hashing variants.

**JV-hashing.** For JV hashing, each row-level score is a PRF following Bernulli(0.5). We plot the empirical probability mass function (PMF) of these scores for both watermarked and unwatermarked tables in Figure 10. As expected, the unwatermarked data yields an approximately symmetric distribution over $\{0, 1\}$, while watermarked tables exhibit a clear shift of probability mass toward larger score values due to multi-sample selection.

**PC-hashing.** For PC hashing, the row-level score is the sum of per-column Bernoulli bits, taking values in $\{0, ..., n\}$ where $n$ is the number of columns. We visualize the empirical PMF over the normalized score (defined in Equation (3)) in Figure 11. Again, unwatermarked tables show the expected symmetric distribution, while watermarked tables exhibit a rightward shift in mass, reflecting the watermark signal.

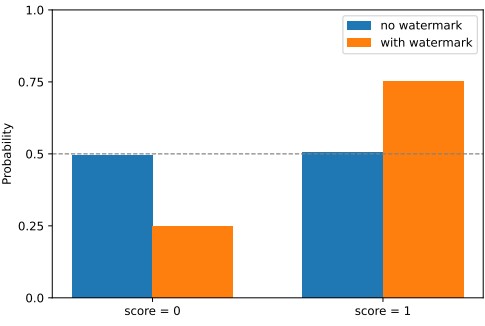

Figure 10: Probability mass function of JV detection score for both watermarked and unwatermarked table. Watermarked table biases the score distribution toward values with larger scores.

Figure 11: Probability mass function of PC detection score for both watermarked and unwatermarked table. Watermarked table biases the score distribution toward values with larger scores.

## D Further Analysis of the Inversion-Based Watermarking

We first introduce the overall pipeline of inversion-based watermarking in Figure 12. The difficulty lies in the inversion of three components, in sequential order: (1) inverse Quantile Transformation (IQT) §D.2, (2) the VAE decoder §D.3, and (3) the DDIM sampling process §D.4. Finally, we analyze the error accumulation and detection performance across the inversion stages in §D.5.

## D.1 PIPELINE OF INVERSION-BASED WATERMARKING

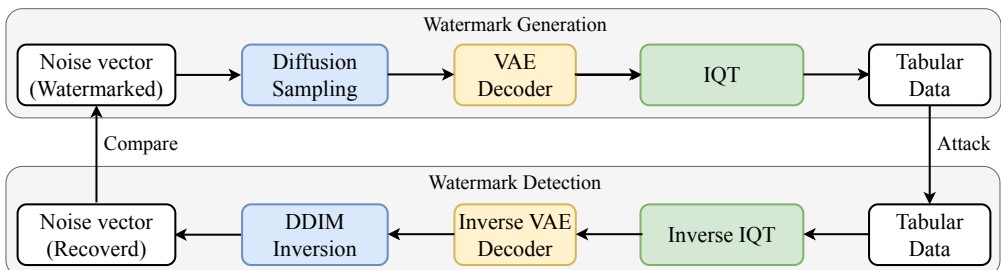

Figure 12: Pipeline of Inversion-based Watermarking. **Top**: The watermark signal is embedded in the noise vector in the latent space, a watermarked table is subsequently generated. **Bottom**: To detect the watermark signal, we need to reverse the entire pipeline. IQT stands for the inverse map of Quantile Transformation.

## D.2 INVERSION OF (INVERSE) QUANTILE TRANSFORMATION

The Quantile Transformation (Wikipedia contributors, 2025) is a widely used (Zhang et al., 2024c;a; Shi et al., 2024; Kotelnikov et al., 2023) data preprocessing step in tabular data synthesis. It regularizes the data distribution to a standard normal distribution. The Quantile Transformation can be implemented as follows:

1) Estimate the empirical cumulative distribution function (CDF) of the features.
2) Map to uniform distribution with the estimated CDF.
3) Map to standard normal distribution with inverse transform sampling: $z = \Phi^{-1}(u)$, where $\Phi$ is the CDF of the standard normal distribution.

Note that in the second step, only the ordering of the data is preserved, and the exact values are not preserved, making the map non-injective, therefore, the inverse of the Quantile Transformation is inherently error-prone. Based on the official codebase, TabWak (Zhu et al., 2025) bypasses the inversion of quantile normalization by caching the original data during watermarking, this is infeasible in practical scenarios where the ground truth is unavailable. To study the impact of the inversion error of the Quantile Transformation, we apply the original Quantile Transformation to the sampled tabular data to invert the inverse quantile transformation.

## D.3 INVERSION OF VAE DECODER

Denote the VAE decoder as $f_\theta$, and the VAE decoder output as $\mathbf{x} = f_\theta(\mathbf{z})$. To get $\mathbf{z}$ from $\mathbf{x}$, (Zhu et al., 2025) employs a gradient-based optimization to approximate the inverse of the VAE decoder. Specifically, we can parametrize the unknown $\mathbf{z}$ with trainable parameters, and optimize the following objective with standard gradient descent:

$$\mathbf{z} = \arg\min_{\mathbf{z}} \|\mathbf{x} - f_\theta(\mathbf{z})\|_2^2.$$

where $\mathbf{z}$ is initaitzed as $g(f_\theta(\mathbf{x}))$, and $g(\cdot)$ is a VAE encoder. However, there is no guarantee that the above optimization will converge to the true $\mathbf{z}$, and we observed that the optimization process is unstable (sometimes produces NaN) for tabular data and introduces significant error in the inversion process.

## D.4 DDIM INVERSION

The DDIM diffusion forward process is defined as:

$$q(\mathbf{x}_t \mid \mathbf{x}_{t-1}) = \mathcal{N}(\mathbf{x}_t; \sqrt{1 - \beta_t}\mathbf{x}_{t-1}, \beta_t\mathbf{I}),$$

where $\mathbf{x}_0$ is the original data, $\mathbf{x}_t$ is the data at time $t$, and $\beta_t$ is the variance of the noise at step $t$. Based on the above definition, we can write $\mathbf{x}_t$ as:

$$\mathbf{x}_t = \sqrt{\bar{\alpha}_t}\mathbf{x}_{t-1} + \sqrt{1 - \bar{\alpha}_t}\epsilon, \tag{Forward process}$$

where $\bar{\alpha}_t = \prod_{i=0}^{t}(1 - \beta_i)$, $\epsilon \sim \mathcal{N}(\mathbf{0}, \mathbf{I})$.

Starting from $\mathbf{x}_T$, we sample $\mathbf{x}_{T-1}, \ldots, \mathbf{x}_0$ recursively according to the following process:

$$\mathbf{x}_0^t = \left(\mathbf{x}_t - \sqrt{1 - \bar{\alpha}_t}\epsilon_\theta(\mathbf{x}_t, t)\right) / \sqrt{\bar{\alpha}_t}$$
$$\mathbf{x}_{t-1} = \sqrt{\bar{\alpha}_{t-1}}\mathbf{x}_0^t + \sqrt{1 - \bar{\alpha}_{t-1}}\epsilon_\theta(\mathbf{x}_t, t), \qquad \text{(Reverse process)}$$

where $\epsilon_\theta(\mathbf{x}_t, t)$ is noise predicted by a neural network.

The **DDIM inversion process** is defined as the inverse of the DDIM reverse process. Specifically, starting from $\mathbf{x}_0$, our goal is to recover the original noise vector $\mathbf{x}_T$ in the latent space. We introduce the basic DDIM inversion process proposed in (Dhariwal and Nichol, 2021), which is widely adopted in inversion-based watermark methods (Wen et al., 2023; Yang et al., 2024; Zhu et al., 2025; Hu et al., 2025).

We can obtain the inverse of the DDIM forward process by replacing the $t-1$ subscript with $t+1$ in Equation (Reverse process), but use $\mathbf{x}_t$ to approximate the unknown $\mathbf{x}_{t+2}$:

$$\mathbf{x}_{t+1} = \sqrt{\bar{\alpha}_{t+1}}\mathbf{x}_0^t + \sqrt{1 - \bar{\alpha}_{t+1}}\epsilon_\theta(\mathbf{x}_t, t),$$

Due to the approximation $\mathbf{x}_t \approx \mathbf{x}_{t+2}$, the inversion process generally demands a finer discretization of the time steps. For instance, inversion-based watermarking methods (Wen et al., 2023; Zhu et al., 2025) typically adopt $T = 1000$ steps, whereas diffusion models optimized for fast inference (Karras et al., 2022; Zhang et al., 2024c) often operate with a coarser discretization of $T = 50$ steps.

**Advanced Inversion Methods.** To address the inexactness of the above inversion process, recent works (Hong et al., 2024; Pan et al., 2023) have proposed more accurate inversion methods based on iterative optimization. However, we empirically found that those methods still suffer from inversion error due to already noisy input from the previous steps (VAE decoder and Quantile Transformation).

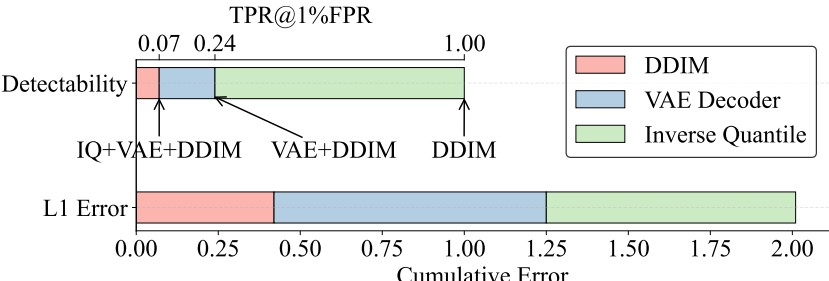

Figure 13: Error Accumulation and Detection Performance Across Inversion Stages of TabWak. The $\ell_1$ error is computed between the estimated and ground truth noise vectors in latent space.

## D.5 ERROR ACCUMULATION

In Figure 13, we analyze the error accumulated at each inversion stage and its impact on detection performance using the `Adult` dataset. Specifically, we compute the TPR@1%FPR over 100 watermarked tables, each with 100 rows. The top bar chart shows detection performance when progressively inverting different parts of the pipeline. From left to right:

- When we invert the entire pipeline (IQ → VAE → DDIM), the detection performance drops to 0.07 TPR@1%FPR.

- When we provide the ground-truth IQ and only invert the VAE decoder and DDIM, the performance improves to 0.24 TPR@1%FPR.

- When both the ground-truth IQ and VAE decoder outputs are provided (i.e., only DDIM is inverted), detection reaches a perfect 1.0 TPR@1%FPR.

The bottom bar chart reports the $\ell_1$ error between the estimated and ground-truth noise vectors in the latent space. From left to right, the bars correspond to:

- Inverting only DDIM (given the ground-truth VAE output),
- Inverting both the VAE decoder and DDIM (given the ground-truth IQ), and
- Inverting the full pipeline (IQ → VAE → DDIM).

This comparison highlights how errors accumulate through the inversion stages and directly affect watermark detectability.

# E   FURTHER ANALYSIS ON ROBUSTNESS

## E.1   COLUMN PERMUTATION ATTACK

In our primary robustness evaluation (Section 5.3), we aligned with prior literature (Zhu et al., 2025) by not explicitly modeling column permutation attacks. This decision relied on the practical assumption that original column ordering is easily recoverable via column headers or statistical properties. However, to evaluate the resilience of our methods under a stricter threat model where column alignment is impossible or headers are stripped, we analyze the impact of column permutation below. We demonstrate that MUSE-PC is naturally robust to this attack, while MUSE-JV can be adapted to achieve permutation robustness with minimal performance trade-offs.

**MUSE-PC.** This variant exhibits inherent invariance to column permutation. Since the watermark detection score for a row is calculated as a summation over all feature columns, the calculation is commutative. Consequently, the spatial arrangement of the columns does not influence the final aggregate score, rendering column permutation attacks ineffective.

**MUSE-JV.** The standard implementation of MUSE-JV relies on pre-computed per-column statistics (see Equation (4)) to determine quantile ranks. A full column permutation disrupts the mapping between columns and their stored statistics. To mitigate this, we can apply a simple modification: estimating the min/max values directly from the target synthetic table rather than relying on pre-stored metadata. This adaptation decouples the detector from specific column indices. While estimating statistics from the sample introduces a potential approximation error compared to the injector's ground truth, our experiments indicate that this deviation is negligible for detection purposes. To validate this, we conducted an experiment where both injection and detection utilized min/max estimates derived from 10,000 independently generated samples. As shown in Table 5, the proposed adaptation maintains high detectability across all datasets.

| Dataset | AUC | T@R 0.1 |
|---|---|---|
| Adult | 1.000 | 1.000 |
| Default | 0.997 | 0.809 |
| Shoppers | 1.000 | 1.000 |
| Beijing | 1.000 | 1.000 |

Table 5: Detection performance under estimated max/min.

## E.2   GLOBAL PERTURBATION

In this section, we extend our evaluation to datasets consisting exclusively of numerical columns. We analyze the performance of MUSE-PC compared to baselines under a threat model where all entries are subject to noise, distinct from the subset perturbation model discussed in the main text.

The original design of MUSE-PC targets a threat model where an adversary perturbs a subset of values with arbitrary strength, while other values remain unchanged. In that regime, robustness is achieved by spreading the watermark signal across all columns. However, in a scenario where every entry is perturbed by small noise (e.g., Gaussian noise), directly computing the score on raw continuous features can be sensitive to these ubiquitous minor shifts.

**Normalization.** To address this, we introduce a lightweight normalization step prior to computing the score. We apply a transformation $f$ such that $f(x) \approx f(x + z)$ when $z$ is a small perturbation. This ensures the downstream score remains stable even if all entries receive noise.

Specifically, we instantiate $f$ as **quantization in the log domain**. The process is as follows:

Table 6: Watermark generation quality and detectability on fully numerical datasets, ▨ indicates best performance, ▨ indicates second-best performance. For clarity, only our method is highlighted in detection.

| Dataset | Method | Watermark Generation Quality | | | | Watermark Detectability | |
|---|---|---|---|---|---|---|---|
| | | Marg.↑ | Corr.↑ | C2ST↑ | MLE↑ | AUC↑ | T@1%F↑ |
| California | no-wm | 0.992 | 0.992 | 0.995 | 0.994 | - | - |
| | TabWak | 0.905 | 0.937 | 0.783 | 0.787 | 0.871 | 0.39 |
| | TabWak* | 0.891 | 0.930 | 0.753 | 0.934 | 0.976 | 0.53 |
| | **MUSE-PC** | 0.933 | 0.964 | 0.851 | 0.994 | 1.000 | 1.00 |
| Letter | no-wm | 0.975 | 0.980 | 0.980 | 0.992 | - | - |
| | TabWak | 0.928 | 0.938 | 0.685 | 0.926 | 0.999 | 0.90 |
| | TabWak* | 0.922 | 0.930 | 0.607 | 0.919 | 1.000 | 1.00 |
| | **MUSE-PC** | 0.928 | 0.964 | 0.740 | 0.990 | 1.000 | 1.00 |

1) Map each numerical value to its logarithmic scale.
2) Assign the value to one of a fixed number of bins (denoted as bin_num).

This logarithmic transformation makes the bin widths adaptive: larger magnitude values ($|x|$) are assigned wider bins. This aligns with the intuition that larger values can tolerate larger absolute perturbations without altering their semantic meaning or watermark bin assignment. This preprocessing does not alter the fundamental sampling or scoring procedure of MUSE-PC.

**Robustness to Global Perturbation.** We evaluate the robustness of MUSE-PC—augmented with a quantisation step prior to score computation—under global perturbations, instantiated as Gaussian noise applied to every entry in fully numerical datasets. We set the number of bins to 32, $N = 500$, $m = 2$, and compare the detectability of MUSE-PC against TabWak and TabWak*. As shown in Figure 14, MUSE-PC consistently outperforms both baselines across all attack strengths on both datasets. It is worth noting that although TabWak/TabWak* demonstrates relatively strong robustness on the `Letter` dataset, its performance deteriorates substantially on `California`. We hypothesize that this variability stems from the inherent instability of reversing the entire sampling pipeline, which TabWak relies on for detection.

**Distortion and Detectability Performance.** While the normalization step renders MUSE-PC robust to global perturbations, evaluating its potential impact on distortion remains critical. With the number of bins fixed at 32, results in Table 6 demonstrate that MUSE-PC consistently outperforms TabWak and TabWak* in terms of both distortion and detectability.

**Ablation Study on Number of Bins.** In this section, we examine the impact of the bin count, $b \in \{16, 32, 64, 128, 256\}$, on the data quality and robustness of MUSE-PC. As shown in Figure 15, Figure 16, and Figure 17, the number of bins introduces a fundamental trade-off: coarser binning enhances robustness at the cost of slightly increased distortion, while finer binning favors fidelity but reduces robustness. Notably, choosing $b = 32$ is sufficient to surpass the robustness of TabWak. Furthermore, even under coarse quantization ($b = 16$), MUSE-PC preserves higher data quality than TabWak.

### E.3 WATERMARK STEALING

In this section, we consider the *watermark stealing* problem, where an adversary attempts to reverse engineer the watermark. We adopt the standard setting under Kerckhoffs' principle: the adversary has full knowledge of the watermarking algorithm, but does not know the secret key.

**Two levels of reverse-engineering.** It is useful to distinguish between two goals an attacker may pursue:

- **Spoofing attack (easier).** The adversary trains a generative model to approximate the watermarked data distribution $P_{\text{wm}}$, with the goal of generating new samples that pass the detector, without necessarily recovering the secret key.

- **Parameter-recovery attack (harder).** The adversary attempts to deduce the secret parameters of the scheme—specifically, the secret key $k$ and/or the exact configuration of the quantile-

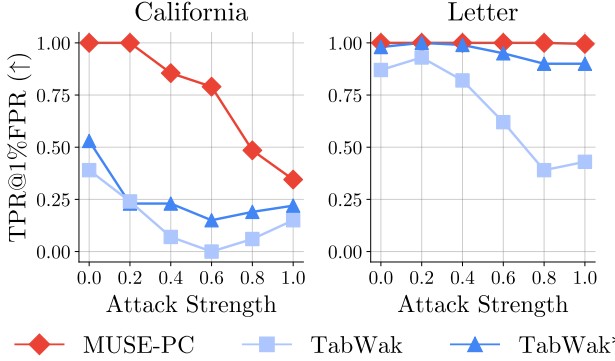

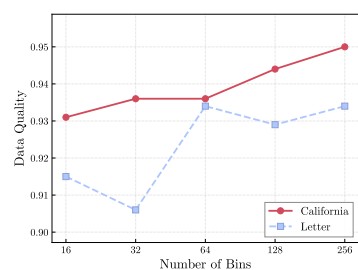

Figure 15: Impact of bin number on data quality. Finer discretization leads to better data quality.

Figure 14: Detection performance under Gaussian perturbation attack across varying attack intensities. MUSE-PC (with number of bin=32) achieves the best robustness.

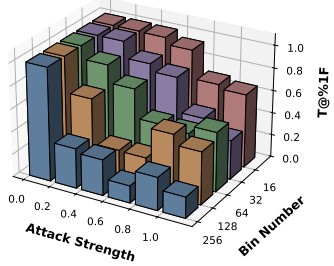

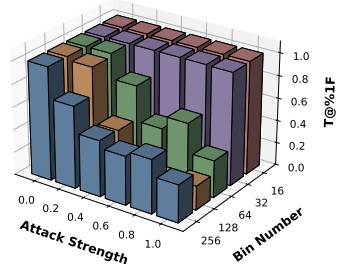

Figure 16: Impact of bin number on robustness to Gaussian perturbation: California dataset.

Figure 17: Impact of bin number on robustness to Gaussian perturbation: Letter dataset.

rank mapping (e.g., which columns and quantile levels are selected and hashed). A successful parameter-recovery attack constitutes a *total break*: once the mechanism (known under Kerckhoffs' principle) and the key are both recovered, the adversary can scrub or spoof the watermark at will.

Parameter recovery is strictly harder than spoofing: if an attacker could recover the key and quantile configuration, they could trivially simulate the watermarking process and thus succeed at spoofing. The converse does not hold: in general, one can statistically approximate a distribution without solving the cryptographic task of key recovery. This mirrors observations in prior watermarking work, where key-recovery attacks are typically bespoke and non-trivial to construct (Jovanović et al., 2024).

**Quantile ranks as a hard parameter-recovery target.** For the sake of simplicity, our JV hashing scheme presented in the main text uses a fixed quantile set (e.g., $\mathcal{Q} = \{0, 0.5, 1\}$ corresponding to minimum, median, and maximum) to select a sparse subset of columns for watermark embedding. Under Kerckhoffs' principle, an attacker would know the fixed quantile set and thus be able to compute which columns are used for watermarking for each sample. We provide a simple security enhancement: applying a keyed pseudorandom permutation (PRP) $\pi_k$ over the column indices before column selection. Specifically, for each row $\mathbf{x}$, we first apply the permutation $\mathbf{x} \mapsto \pi_k(\mathbf{x})$, and then compute the quantile ranks and select the minimum/median/maximum positions in this permuted order. Under this construction, the set of watermark-carrying columns is entirely determined by the secret key $k$, and recovering it is computationally equivalent to inverting the underlying PRP. In other words, reverse-engineering the quantile-rank configuration becomes a full-fledged parameter-recovery attack on a cryptographic primitive, which is significantly harder than merely mimicking the watermark's statistical footprint.

**Empirical Evidence.** Our analysis is supported empirically by the *Adaptive Adversary* experiments presented in Section 5.3. We simulated a distillation attack—representing the easier "Spoofing" threat—where a powerful generative model (TabSyn) attempted to learn the watermarked distribution. As shown in Figure 4, the adversary failed to distinguish or approximate the watermark signal (achieving an AUC $\approx 0.5$). Since the adversary failed at the easier task of statistical approximation (Spoofing), we conclude that they statistically cannot succeed at the strictly harder task of Parameter Recovery.

## F  EXPERIMENTAL DETAILS

### F.1  HARDWARE SPECIFICATION

We use a single hardware for all experiments. The hardware specifications are as follows:

- GPU: NVIDIA RTX 4090
- CPU: Intel 14900K

### F.2  DATASET STATISTICS

The dataset used in this paper could be automatically downloaded using the script in the provided code. We use 6 tabular datasets from UCI Machine Learning Repository[1] or Kaggle[2]: Adult[3], Default[4], Shoppers[5], Beijing[6], California[7], and Letter[8], which contain different numbers of numerical and categorical features. The statistics of the datasets are presented in Table 7.

Table 7: Dataset statistics.

| **Dataset** | # Rows | # Continuous | # Discrete | # Target | # Train | # Test | Task |
|---|---|---|---|---|---|---|---|
| **Adult** | $32,561$ | 6 | 8 | 1 | $22,792$ | $16,281$ | Classification |
| **Default** | $30,000$ | 14 | 10 | 1 | $27,000$ | $3,000$ | Classification |
| **Shoppers** | $12,330$ | 10 | 7 | 1 | $11,098$ | $1,232$ | Classification |
| **Beijing** | $43,824$ | 7 | 5 | 1 | $39,441$ | $4,383$ | Regression |
| **California** | $20,640$ | 9 | - | 1 | $18,390$ | $2,520$ | Classification |
| **Letter** | $20,000$ | 16 | - | 1 | $18,000$ | $2,000$ | Classification |

In Table 7, **# Rows** refers to the total records in each dataset, while **# Continuous** and **# Discrete** denote the count of numerical and categorical features, respectively. The **# Target** column indicates whether the prediction task involves a continuous (regression) or discrete (classification) target variable. All datasets except Adult are partitioned into training and testing sets using a 9:1 ratio, with splits generated using a fixed random seed for reproducibility. The Adult dataset uses its predefined official testing set. For evaluating Machine Learning Efficiency (MLE), the training data is further subdivided into training and validation subsets with an 8:1 ratio, ensuring consistent evaluation protocols across experiments.

### F.3  FIDELITY METRICS

The fidelity metrics used in this paper (Marginal, Correlation, C2ST and MLE) are standard metrics in the field of tabualr data synthesis. Here is a reference:

- Marginal: Appendix E.3.1 in (Zhang et al., 2024c).

---

[1] https://archive.ics.uci.edu/datasets
[2] https://www.kaggle.com
[3] https://archive.ics.uci.edu/dataset/2/adult
[4] https://archive.ics.uci.edu/dataset/350/default+of+credit+card+clients
[5] https://archive.ics.uci.edu/dataset/468/online+shoppers+purchasing+intention+dataset
[6] https://archive.ics.uci.edu/dataset/381/beijing+pm2+5+data
[7] https://www.kaggle.com/datasets/camnugent/california-housing-prices
[8] https://archive.ics.uci.edu/dataset/59/letter+recognition

- Correlation: Appendix E.3.2 in (Zhang et al., 2024c).
- C2ST: Appendix F.3 in (Zhang et al., 2024c).
- MLE: Appendix E.4 in (Zhang et al., 2024c).

Below is a summary of how these metrics work.

### F.3.1 MARGINAL DISTRIBUTION

The **Marginal** metric assesses how well the marginal distribution of each column is preserved in the synthetic data. For continuous columns, we use the Kolmogorov–Smirnov Test (KST); for categorical columns, we use the Total Variation Distance (TVD).

**Kolmogorov–Smirnov Test (KST)**    Given two continuous distributions $p_r(x)$ and $p_s(x)$ (real and synthetic, respectively), the KST measures the maximum discrepancy between their cumulative distribution functions (CDFs):

$$\text{KST} = \sup_x |F_r(x) - F_s(x)|,\tag{8}$$

where $F_r(x)$ and $F_s(x)$ denote the CDFs of $p_r(x)$ and $p_s(x)$:

$$F(x) = \int_{-\infty}^{x} p(x)\, \mathrm{d}x.\tag{9}$$

**Total Variation Distance (TVD)**    TVD measures the difference between the categorical distributions of real and synthetic data. Let $\Omega$ be the set of possible categories in a column. Then:

$$\text{TVD} = \frac{1}{2} \sum_{\omega \in \Omega} |R(\omega) - S(\omega)|,\tag{10}$$

where $R(\cdot)$ and $S(\cdot)$ denote the empirical probabilities in real and synthetic data, respectively.

### F.3.2 CORRELATION

The **Correlation** metric evaluates whether pairwise relationships between columns are preserved.

**Pearson Correlation Coefficient**    For two continuous columns $x$ and $y$, the Pearson correlation coefficient is defined as:

$$\rho_{x,y} = \frac{\text{Cov}(x,y)}{\sigma_x \sigma_y},\tag{11}$$

where $\text{Cov}(\cdot)$ is the covariance and $\sigma$ denotes standard deviation. We evaluate the preservation of correlation by computing the mean absolute difference between correlations in real and synthetic data:

$$\text{Pearson Score} = \frac{1}{2}\mathbb{E}_{x,y} \left| \rho^R(x,y) - \rho^S(x,y) \right|,\tag{12}$$

where $\rho^R$ and $\rho^S$ denote correlations in real and synthetic data. The score is scaled by $\frac{1}{2}$ to ensure it lies in $[0,1]$. Lower values indicate better alignment.

**Contingency Similarity**    For categorical columns $A$ and $B$, we compute the Total Variation Distance between their contingency tables:

$$\text{Contingency Score} = \frac{1}{2} \sum_{\alpha \in A} \sum_{\beta \in B} |R_{\alpha,\beta} - S_{\alpha,\beta}|,\tag{13}$$

where $R_{\alpha,\beta}$ and $S_{\alpha,\beta}$ are the joint frequencies of $(\alpha, \beta)$ in the real and synthetic data, respectively.

### F.3.3 CLASSIFIER TWO-SAMPLE TEST (C2ST)

C2ST evaluates how distinguishable the synthetic data is from real data. If a classifier can easily separate the two, the synthetic data poorly approximates the real distribution. We adopt the implementation provided by the SDMetrics library.[9]

### F.3.4 MACHINE LEARNING EFFICIENCY (MLE)

MLE evaluates the utility of synthetic data for downstream machine learning tasks. Each dataset is split into training and testing subsets using real data. Generative models are trained on the real training set, and a synthetic dataset of equal size is sampled.

For both real and synthetic data, we use the following protocol:

- Split the training set into train/validation with an $8:1$ ratio.
- Train a classifier/regressor on the train split.
- Tune hyperparameters based on validation performance.
- Retrain the model on the full training set using the optimal hyperparameters.
- Evaluate on the real test set.

This process is repeated over 20 random train/validation splits. Final scores (AUC for classification task or RMSE for regression task) are averaged over the 20 trials for both real and synthetic training data. In our experiments, we report the MLE Gap, which is the difference between the MLE score of the (unwatermarked) real data and the MLE score of the synthetic data.

### F.4 WATERMARK DETECTION METRICS

For watermark detection metrics, we primarily use the area under the curve (AUC) of the receiver operating characteristic (ROC) curve: **AUC**, and the True Positive Rate (TPR) at a given False Positive Rate (FPR): **TPR@x%FPR**.

$z$**-statistic** In addition, we can formalize a statistical test for watermark detection. We formulate this as a hypothesis testing problem:

$$H_0 : \text{The table is not watermarked.}$$
$$\text{vs. } H_1 : \text{The table is watermarked.}$$

Recall the definition of our detection statistic in Equation (1): given a (watermarked or unwatermarked) table $T$ that consists of $N$ rows: $T := (\mathbf{x}_1, \ldots, \mathbf{x}_N)$, we compute the detection statistic:

$$S(T) = \frac{1}{N} \sum_{i=1}^{N} s_k(\mathbf{x}_i).$$

For the Joint-Vector (JV) hashing design, where each row is assigned a single score, the form of the test statistic depends on the score's distribution under the null hypothesis $H_0$. If the row score follows Burnulli(0.5), we denote the total count of rows with a score of 1 as $|W|$. Under $H_0$, $|W|$ follows a binomial distribution with mean $\mu = N/2$ and variance $\sigma^2 = N/4$. Finally, since $|W| = \sum_{i=1}^{N} s_k(\mathbf{x}_i) = N \cdot S(T)$, thus the $z$-statistic is computed as:

$$z = \frac{N \cdot S(T) - N/2}{\sqrt{N/4}}. \qquad \text{(JV hash)}$$

For the Per-Column (PC) design, this framework must be adapted, as the score for each row, $s_i$, is the average of scores from $M$ individual columns (see Equation (3)): $s_k(\mathbf{x}) = \frac{1}{M} \sum_{j=1}^{M} C_{ij}$,

---

[9]https://docs.sdv.dev/sdmetrics/metrics/metrics-in-beta/
detection-single-table

where $C_{ij}$ is the score assigned to the value at $i$-th row, $j$-th column. If $C_{i,j}$ are $i.i.d$ and follows Bernulli(0.5), we have that $\sum_{i=1}^{N} s_k(\mathbf{x}_i)$ follows a binomial distribution with mean $\mu = \frac{M \cdot N}{2}$ and variance $\sigma^2 = \frac{N \cdot M}{4}$, yielding a $z-$statistic as follows:

$$z = \frac{N \cdot S(T) - \frac{N \cdot M}{2}}{\sqrt{\frac{N \cdot M}{4}}}. \tag{JV hash}$$

**Estimating the statistic under $H_0$ via Monte Carlo.** While directly assuming certain distributions under null hypotheses like above is standard in LLM watermarking (Kirchenbauer et al., 2023b; Zhao et al., 2023a; Giboulot and Furon, 2024), they can be inaccurate when the table contains low-entropy columns (e.g., binary attributes). In such cases, the exact distribution of row-level hash outputs under $H_0$ may deviate from the idealized Bernoulli model.

One way to address this problem is to estimate the mean and variance of the detection statistic under $H_0$ using *Monte Carlo simulation*, which is also used in TabWak (Zhu et al., 2025).

Specifically, we first sample $K$ unwatermarked tables with $N$ rows, denoted as $T_{nw}^1, ..., T_{nw}^K$. Denote $s_i$ as $N \cdot S(T_{nw}^i)$, then we compute:

- $\hat{\mu}_{nw}$: the empirical mean of $\{s_1, ..., s_K\}$.
- $\hat{\sigma}_{nw}$: the empirical standard derivation of $\{s_1, ..., s_K\}$.

Then the one-sided $z$-statistic can be computed as:

$$z = \frac{N \cdot S(T) - \hat{\mu}_{nw}}{\hat{\sigma}_{nw}}$$

where $s$ is the test statistic computed on the suspect table. Unlike TabWak (Zhu et al., 2025), no additional $1/\sqrt{N}$ scaling is required because $\hat{\sigma}_{nw}$ is estimated directly from the full statistic $N \cdot S(T)$, whose variance already incorporates the dependence on $N$.

**Detection threshold.** Given the estimated (or assumed) null distribution of our detection statistic, we next define a threshold for deciding whether a table is watermarked. Let $\mu_0$ and $\sigma_0$ denote the mean and standard deviation of the statistic under $H_0$, obtained either analytically (e.g., assuming a Bernoulli or binomial model) or empirically via Monte Carlo simulation as described above. For a suspect table $T$, the corresponding $z$-score is:

$$z = \frac{N \cdot S(T) - \mu_0}{\sigma_0}. \tag{14}$$

To control the false-positive rate at a user-specified significance level $\alpha$, we compute the critical value $z_\alpha$ such that

$$\Pr(Z > z_\alpha \mid H_0) = \alpha, \qquad Z \sim \mathcal{N}(0,1), \tag{15}$$

and declare the table as watermarked whenever $z > z_\alpha$. Equivalently, this induces a threshold on the normalized statistic $S(T)$:

$$S(T) > \frac{\mu_0}{N} + z_\alpha \frac{\sigma_0}{N}. \tag{16}$$

When the theoretical Bernoulli(0.5) assumption holds (e.g., JV-hash), we have $\mu_0 = N/2$ and $\sigma_0 = \sqrt{N/4}$, which recovers the familiar closed-form thresholds used in prior work. When Monte Carlo estimation is used instead, the same decision rule applies but with empirical estimates $(\hat{\mu}_{nw}, \hat{\sigma}_{nw})$, enabling the threshold to automatically adapt to low-entropy or skewed tabular datasets.

## F.5 IMPLEMENTATION DETAILS OF IMAGE WATERMARK BASELINES

In this work, we also benchmark our method against established watermarking techniques originally designed for visual generative models: Tree-Ring Watermark (Wen et al., 2023) and Gaussian Shading (Yang et al., 2024). To apply these image-based methods to the tabular domain, we strictly follow the adaptation strategies proposed in TabWak (Zhu et al., 2025). We include a brief description of these strategies below for completeness; for full algorithmic details, we refer readers to Appendix D of TabWak.

**Tree-Ring Watermark.** This method embeds the watermark into the initial noise vector of the diffusion process by transforming it into the frequency domain. Importantly, this method treats the full table ($m$ rows and $n$ columns) as a single latent image for the watermark. While standard image models typically process square inputs, where standard centralized ring patterns are embedded in the latent. Tabular datasets are characterized by a high aspect ratio, where the number of rows ($m$) significantly exceeds the number of columns ($n$). To address this geometric discrepancy, we embed a *ripple-shaped pattern* across the Fourier space. However, it is worth noting that treating the full table as a single unit makes this method inherently vulnerable to row shuffling attacks: simply permuting the rows destroys the global spatial pattern, thereby severely compromising detectability.

**Gaussian Shading.** Unlike the Tree-Ring watermark, Gaussian Shading is applied at the individual row level. This approach treats each tabular row as a distinct entity, similar to how watermarking is applied to individual images. Crucially, we maintain a fixed control seed across the entire dataset. If we were to assign a unique seed to each row index, a simple row shuffling attack would decouple the data from its corresponding seed, making verification impossible. By enforcing a constant seed, we ensure that the watermark remains detectable even if the rows are arbitrarily permuted.

**Discussion.** The key distinction between watermarking techniques for tabular data and those for images lies in the application setting. Tabular watermarking typically operates on an entire table—a batch of i.i.d. samples—where each row contributes to the aggregate $z$-score and collectively boosts detectability. In contrast, image watermarking generally requires detecting a watermark from a *single* generated instance. For example, as shown in Theorem 4.1, achieving a target detectability of FPR = 0.01% under MUSE requires a batch size of $N = 100$ and $m = 4$ repeated samples per instance. Applied to images, this would require roughly 400 forward passes of an image generator to watermark a batch of 100 images, making the method impractical for standard single-image watermarking scenarios. That said, in specialized applications where images are naturally generated and verified in batches, MUSE could still offer a viable and effective watermarking strategy.

## G OMMITED PROOFS IN SECTION 3

Recall that for a table $T$ (wateramarked or unwatermarked) with $N$ rows: $\mathbf{x}_1, \ldots, \mathbf{x}_N$, we define the watermark detection score as

$$S(T) = \frac{1}{N} \sum_{i=1}^{N} s_k(\mathbf{x}_i), \tag{17}$$

where $s_k(\mathbf{x}_i)$ is the score of the $i$-th sample, $k$ is the fixed watermark key.

**Theorem 4.1** (Watermark Calibration Guarantees). *Denote a watermarked table as $T_{\text{wm}}$ and an unwatermarked table as $T_{\text{no-wm}}$, each consisting of $N$ rows. Let $\mathbf{x} \sim p(\mathbf{x})$ be a random variable drawn from the data distribution, and let $\mathbf{x}_1, \ldots, \mathbf{x}_m$ be i.i.d. samples from $p(\mathbf{x})$. Define $\mu_{\text{no-wm}} = \mathbb{E}_{\mathbf{x} \sim p(\mathbf{x})}[s_k(\mathbf{x})]$ as the expected score of an unwatermarked sample, and define $\mu_{\text{wm}}^m = \mathbb{E}_{\mathbf{x}_i \sim p(\mathbf{x})} \left[ \max_{i \in [m]} s_k(\mathbf{x}_i) \right]$ as the expected score of a watermarked sample obtained via $m$ repeated samples. Suppose the scoring function satisfies $s_k(\cdot) \in [0, 1]$, we have:*

1. *The False Positive Rate (FPR) of the watermark detection is upper bounded:*

$$\Pr\left(S(T_{\text{no-wm}}) > S(T_{\text{wm}})\right) \leq \exp\left(-\frac{N \cdot (\mu_{\text{wm}}^m - \mu_{\text{no-wm}})^2}{2}\right). \tag{5}$$

2. *The RHS of the bound is minimized when $s_k(\mathbf{x})$ follows a $\text{Bernoulli}(0.5)$ distribution.*

3. *Under this optimal distribution, let $N > 8\log(1/\alpha)$, then to ensure the FPR does not exceed a target threshold $\alpha$, it suffices to set the number of repeated samples $m$ as:*

$$m = \max\left(2, \left\lceil \log_{0.5}\left(0.5 - \sqrt{\tfrac{2\log(1/\alpha)}{N}}\right) \right\rceil\right). \tag{6}$$

*Proof.* The proof of each statement is provided in Lemma G.1, Lemma G.2, and Theorem G.3, respectively. □

**Lemma G.1.** *Denote a watermarked table as $T_{\text{wm}}$ and an unwatermarked table as $T_{\text{no-wm}}$, each consisting of $N$ rows. Let $\mathbf{x} \sim p(\mathbf{x})$ be a random variable drawn from the data distribution, and let $\mathbf{x}_1, \ldots, \mathbf{x}_m$ be i.i.d. samples from $p(\mathbf{x})$. Define $\mu_{\text{no-wm}} = \mathbb{E}_{\mathbf{x} \sim p(\mathbf{x})}[s_k(\mathbf{x})]$ as the expected score of an unwatermarked sample, and define $\mu_{\text{wm}}^m = \mathbb{E}_{\mathbf{x}_i \sim p(\mathbf{x})}\left[\max_{i \in [m]} s_k(\mathbf{x}_i)\right]$ as the expected score of a watermarked sample obtained via $m$ repeated samples. Suppose the scoring function satisfies $s_k(\cdot) \in [0, 1]$, then the False Positive Rate (FPR) of the watermark detection satisfies:*

$$\Pr\left(S(T_{\text{no-wm}}) > S(T_{\text{wm}})\right) \leq \exp\left(-\frac{N(\mu_{\text{wm}}^m - \mu_{\text{no-wm}})^2}{2}\right). \tag{18}$$

*Proof.* Let $S(T_{\text{no-wm}}) = \sum_{i=1}^{N} c_i$ denote the sum of $N$ i.i.d. scores from the unwatermarked table, where each $c_i = s_k(\mathbf{x}_i)$ for $\mathbf{x}_i \sim p(\mathbf{x})$, and similarly let $S(T_{\text{wm}}) = \sum_{i=1}^{N} c_i'$ denote the sum of $N$ i.i.d. scores from the watermarked table, where each $c_i' = \max\{s_k(\mathbf{x}_{i1}), \ldots, s_k(\mathbf{x}_{im})\}$ with $\mathbf{x}_{ij} \sim p(\mathbf{x})$.

Define the expected values:

$$\mu_{\text{no-wm}} = \mathbb{E}[c_i], \quad \mu_{\text{wm}}^m = \mathbb{E}[c_i'].$$

We are interested in bounding the false positive rate:

$$\Pr(S(T_{\text{no-wm}}) > S(T_{\text{wm}})) = \Pr\left(\sum_{i=1}^{N}(c_i - c_i') > 0\right).$$

Let $w_i = c_i - c_i'$. Since $s_k(x) \in [0, 1]$, we have $c_i \in [0, 1]$ and $c_i' \in [0, 1]$, so $w_i \in [-1, 1]$. Moreover, $\mathbb{E}[w_i] = \mu_{\text{no-wm}} - \mu_{\text{wm}}^m =: -\delta$, where $\delta = \mu_{\text{wm}}^m - \mu_{\text{no-wm}} > 0$.

We apply Hoeffding's inequality to the sum of $w_i$'s:

$$\Pr\left(\sum_{i=1}^{N} w_i > 0\right) = \Pr\left(\sum_{i=1}^{N} w_i - \mathbb{E}[\sum_{i=1}^{N} w_i] > N\delta\right) \leq \exp\left(-\frac{2N^2\delta^2}{4N}\right).$$

Plug in the definition of $\delta$, we have:

$$\Pr(S(T_{\text{no-wm}}) > S(T_{\text{wm}})) \leq \exp\left(-\frac{N^2\delta^2}{2}\right) = \exp\left(-\frac{N(\mu_{\text{wm}}^m - \mu_{\text{no-wm}})^2}{2}\right).$$

which proves the result. $\qquad \square$

**Lemma G.2** (Optimal Scoring Distribution). *Let $s_k(\mathbf{x})$ be any random variable supported on $[0, 1]$ with mean $0.5$, the right-hand-side of Equation (18) is minimized when $s_k(\mathbf{x})$ follows a Bernoulli$(0.5)$ distribution.*

*Proof.* Let $s_1, \ldots, s_m$ be i.i.d. copies of a random variable $s_k(\mathbf{x}) \in [0, 1]$ with fixed mean $\mathbb{E}[s_k(\mathbf{x})] = 0.5$. Define:

$$\mu := \mathbb{E}[s_k(\mathbf{x})] = 0.5, \quad \mu_{\max} := \mathbb{E}[\max(s_1, \ldots, s_m)].$$

Let $\Delta := \mu_{\max} - \mu$ be the gap between the expected maximum score over $m$ repetitions and the mean score. The upper bound in Equation (18) is:

$$\Pr(S_{\text{no-wm}} > S_{\text{wm}}) \leq \exp\left(-\frac{N\Delta^2}{2}\right),$$

so minimizing the FPR corresponds to maximizing $\Delta$ under the constraint that $\mathbb{E}[s_k(\mathbf{x})] = 0.5$ and $s_k(\mathbf{x}) \in [0, 1]$.

We now show that $\Delta$ is maximized when $s_k(\mathbf{x}) \sim \text{Bernoulli}(0.5)$.

**Step 1: Write $\mu_{\max}$ and $\mu$ as integrals over the CDF.** Let $F$ be the cumulative distribution function (CDF) of $s_k(\mathbf{x})$. Then the CDF of $\max(s_1, \ldots, s_m)$ is $F^m(x)$. By the tail integration formula, we can compute the expected maximum as:

$$\mu_{\max} = \int_0^1 \Pr(\max(s_1, \ldots, s_m) > x)$$

$$= \int_0^1 (1 - F(x)^m)\, dx.$$

Similarly, we have: $\mu = \int_0^1 (1 - F(x))\, dx$.

Therefore, the gap $\Delta$ can be written as:

$$\Delta = \mu_{\max} - \mu = \int_0^1 [F(x) - F(x)^m]\, dx.$$

**Step 2: Leverage the concavity.** By Lemma H.1, the integrand $F(x) - F(x)^m$ is concave in $F(x)$. By Lemma H.2, the integral is maximized when $F(x)$ is the CDF of a Bernoulli distribution with mean $\mu = 0.5$.

Therefore, among all $s_k(\mathbf{x}) \in [0, 1]$ with $\mathbb{E}[s_k(\mathbf{x})] = 0.5$, the Bernoulli$(0.5)$ distribution maximizes $\Delta$, which minimizes the upper bound on the FPR. Hence, the lemma holds. $\qquad\square$

**Theorem G.3** (Minimum Watermarking Signal). *Under the same assumptions as in Lemma G.1, suppose the scoring function $s_k(\mathbf{x})$ is instantiated as a hash-seeded pseudorandom function such that $s_k(\mathbf{x}) \sim Bernoulli(0.5)$. Then the FPR is upper-bounded by:*

$$\Pr\left(S(T_{\text{no-wm}}) > S(T_{\text{wm}})\right) \leq \exp\left(-\frac{N}{2}(0.5 - 0.5^m)^2\right). \tag{19}$$

*To ensure the FPR does not exceed a target threshold $\alpha$, it suffices to set the number of repeated samples $m$ as:*

$$m = \max\left(2, \left\lceil \log_{0.5}\left(0.5 - \sqrt{\tfrac{2\log(1/\alpha)}{N}}\right)\right\rceil\right), \tag{20}$$

*where $\lceil \cdot \rceil$ denotes the ceiling function. This expression is valid when $N > 8\log(1/\alpha)$.*

*Proof.* When $s_k(\mathbf{x}) \sim \text{Bernoulli}(0.5)$, we have:

$$\mu_{\text{no-wm}} = \mathbb{E}[s_k(\mathbf{x})] = 0.5, \quad \mu_{\text{wm}}^m = \mathbb{E}[\max(s_1, \ldots, s_m)] = 1 - 0.5^m.$$

Plug in into the FPR bound Equation (22), we have:

$$\Pr\left(S(T_{\text{no-wm}}) > S(T_{\text{wm}})\right) \leq \exp\left(-\frac{N}{2}(0.5 - 0.5^m)^2\right),$$

which completes the proof. $\qquad\square$

**Theorem 4.3.** *Let $m = 2$. The watermarking process in Algorithm 1, augmented with repeated column masking, satisfies multi-sample distribution-preserving as defined in Definition 4.2.*

*Proof.* Suppose $\tilde{\mathbf{x}}_1, \ldots, \tilde{\mathbf{x}}_K$ are generated consecutively from Algorithm 1 with the same watermark key $k$ and data distribution $p(\mathbf{x})$. Assume the repeated column masking is enabled. Denote $W \subseteq \{1, ..., K\}$ denote the index set where the repeated column masking is triggered. Then we have:

$$\mathbb{P}(\tilde{\mathbf{x}}_1, \ldots, \tilde{\mathbf{x}}_K) = \prod_{i=1}^K \mathbb{P}(\tilde{\mathbf{x}}_i \mid \tilde{\mathbf{x}}_{<i})$$

$$= \underbrace{\prod_{i \in W} \mathbb{P}(\tilde{\mathbf{x}}_i \mid \tilde{\mathbf{x}}_{<i})}_{①} \underbrace{\prod_{i \notin W} \mathbb{P}(\tilde{\mathbf{x}}_i \mid \tilde{\mathbf{x}}_{<i})}_{②}$$

Due to the deployment of repeated column masking, when repeated column values are detected, Algorithm 1 defaults to skipping the watermarking process. Therefore, for ①, we have:

$$\prod_{i \in W} \mathbb{P}(\tilde{\mathbf{x}}_i \mid \tilde{\mathbf{x}}_{<i}) = \prod_{i \in W} p(\tilde{\mathbf{x}}_i)$$

For ②, there will be no repeated column values used for seed generation. Note the dependency between current sample $\mathbf{x}_i$ and previous samples $\tilde{\mathbf{x}}_{<i}$ are only on the watermark key $k$ and selected column values $\pi(\mathbf{x})$ (recall we compute a hash function $h(k, \pi(\mathbf{x}))$ to seed a score function). Therefore, when the selected columns contain no repeated values, due to the property of the hash function, we have $\tilde{\mathbf{x}}_i$ is independent of $\tilde{\mathbf{x}}_{<i}$. Therefore, we have:

$$\prod_{i \notin W} \mathbb{P}(\tilde{\mathbf{x}}_i \mid \tilde{\mathbf{x}}_{<i}) = \prod_{i \notin W} \mathbb{P}(\Gamma(p, h(k, \tilde{\mathbf{x}}_i)))$$
$$= \prod_{i \notin W} p(\tilde{\mathbf{x}}_i) \qquad \text{(by Lemma H.3)}$$

Finally, we combine the above results, we have:

$$\mathbb{P}(\tilde{\mathbf{x}}_1, \ldots, \tilde{\mathbf{x}}_K) = \prod_{i=1}^{K} p(\tilde{\mathbf{x}}_i)$$

which completes the proof. □

## H    TECHNICAL LEMMAS

**Lemma H.1.** *For any integer $m \geq 2$, the function $f(x) = x - x^m$ is concave on the interval $[0, 1]$.*

*Proof.* To prove that $f(x) = x - x^m$ is concave on $[0, 1]$, we show that its second derivative is non-positive on this interval.

Compute the first derivative:

$$f'(x) = \frac{d}{dx}(x - x^m) = 1 - mx^{m-1}.$$

Compute the second derivative:

$$f''(x) = \frac{d}{dx}(1 - mx^{m-1}) = -m(m-1)x^{m-2}.$$

Observe that for all $x \in [0, 1]$ and $m \geq 2$: $m(m-1) > 0$ and $x^{m-2} \geq 0$.

Therefore,
$$f''(x) = -m(m-1)x^{m-2} \leq 0 \quad \text{for all } x \in [0, 1].$$

Hence, $f(x)$ is concave on $[0, 1]$. □

**Lemma H.2.** *Let $\phi : [0, 1] \to \mathbb{R}$ be a concave function, and let $F$ be the cumulative distribution function (CDF) of a random variable supported on $[0, 1]$ with fixed mean $\mu \in (0, 1)$. Then the integral*

$$\int_0^1 \phi(F(x)) dx$$

*is maximized when* $F(x) = \begin{cases} 0 & \text{if } x < 0 \\ 1 - \mu & \text{if } 0 \leq x < 1 \\ 1 & \text{if } x \geq 1 \end{cases}$, *i.e. the CDF of a Bernoulli distribution with mean $\mu$.*

*Proof.* **Step 1: Rewrite the Mean Constraint**

By the tail integration formula, the mean constraint for the random variable $X$ with CDF $F(x)$ supported on $[0, 1]$ is:

$$\int_0^1 (1 - F(x)) \, dx = \mu.$$

Rearranging this equation gives the integral of $F(x)$:

$$\int_0^1 F(x) \, dx = 1 - \mu. \tag{21}$$

**Step 2: Upper Bound the Integral**

The function $\phi : [0, 1] \to \mathbb{R}$ is concave. The CDF $F(x)$ takes values in $[0, 1]$ for $x \in [0, 1]$, so $\phi(F(x))$ is well-defined. We can apply Jensen's inequality for integrals, which for a concave function $\phi$ and an integrable function $g(x)$ on an interval $[a, b]$ states:

$$\frac{1}{b - a} \int_a^b \phi(g(x)) \, dx \leq \phi \left( \frac{1}{b - a} \int_a^b g(x) \, dx \right).$$

Plug in $a = 0$, $b = 1$, $g(x) = F(x)$. Jensen's inequality then becomes:

$$\int_0^1 \phi(F(x)) \, dx \leq \phi \left( \int_0^1 F(x) \, dx \right).$$

Substituting Equation (21) into the right hand side, we have:

$$\int_0^1 \phi(F(x)) \, dx \leq \phi(1 - \mu). \tag{22}$$

**Step 3: Verify $F(x)$ achieves the upper bound**

It is straightforward to verify that $F(x)$ satisfies the mean constraint. Next, we will show that $F(x)$ achieves the upper bound $\phi(1 - \mu)$. For $x \in [0, 1)$, $F(x) = 1 - \mu$. Therefore, we have:

$$\int_0^1 \phi(F(x)) \, dx = \int_0^1 \phi(1 - \mu) \, dx = \phi(1 - \mu).$$

We have shown that $F(x)$ satisfies the mean constraint and achieves the upper bound $\phi(1 - \mu)$, which completes the proof. $\square$

The following proof adapts the single-token distortion-free analysis from (Dathathri et al., 2024) to our single-sample setting. The core ideas and structure of the proof remain the same, with modifications primarily to the notation.

**Lemma H.3** (Single Sample Distortion-free). *Assume $m = 2$, for any data distribution $p(\cdot)$, it holds that, under the randomness of the watermark key $k$, the watermarked data distribution is the same as the original data distribution:*

$$\mathbb{P}_{k \sim \text{Unif}(\mathcal{K})}(\Gamma(p, k) = \tilde{\mathbf{x}}) = p(\tilde{\mathbf{x}}) \tag{23}$$

*Proof.* By definition of the watermarking mechanism with $m = 2$, for any sample $\tilde{\mathbf{x}}$ we can write

$$\mathbb{P}_{k \sim \text{Unif}(\mathcal{K})}(\Gamma(p, k) = \tilde{\mathbf{x}})$$

$$= \mathbb{E}_{k \sim \text{Unif}(\mathcal{K})} \left[ p(\tilde{\mathbf{x}}) \left( \sum_{\mathbf{x} \in \mathcal{X} : s_k(\mathbf{x}) = s_k(\tilde{\mathbf{x}})} p(\mathbf{x}) + 2 \sum_{\mathbf{x} \in \mathcal{X} : s_k(\mathbf{x}) < s_k(\tilde{\mathbf{x}})} p(\mathbf{x}) \right) \right]$$

$$= \mathbb{E}_{k \sim \text{Unif}(\mathcal{K})} \left[ p(\tilde{\mathbf{x}}) \left( \sum_{\mathbf{x} \in \mathcal{X}} p(\mathbf{x}) \left[ \mathbf{1}_{s_k(\mathbf{x}) = s_k(\tilde{\mathbf{x}})} + 2 \mathbf{1}_{s_k(\mathbf{x}) < s_k(\tilde{\mathbf{x}})} \right] \right) \right]$$

where $s_k(\mathbf{x})$ is the score function on sample $\mathbf{x}$ with key $k$.

Next observe that for any fixed $\mathbf{x}$, under $k \sim \mathrm{Unif}(\mathcal{K})$ we have:

$$\mathbb{E}_{k\sim\mathrm{Unif}(\mathcal{K})}\left[\mathbf{1}_{(s_k(\mathbf{x}),k)=s_k(\tilde{\mathbf{x}})} + 2\mathbf{1}_{s_k(\mathbf{x})<s_k(\tilde{\mathbf{x}})}\right]$$
$$= \mathbb{E}_{k\sim\mathrm{Unif}(\mathcal{K})}\left[\mathbf{1}_{(s_k(\mathbf{x}),k)=s_k(\tilde{\mathbf{x}})}\right] + \mathbb{E}_{k\sim\mathrm{Unif}(\mathcal{K})}\left[\mathbf{1}_{s_k(\mathbf{x})<s_k(\tilde{\mathbf{x}})}\right] + \mathbb{E}_{k\sim\mathrm{Unif}(\mathcal{K})}\left[\mathbf{1}_{s_k(\mathbf{x})>s_k(\tilde{\mathbf{x}})}\right]$$
$$= \mathbb{E}_{k\sim\mathrm{Unif}(\mathcal{K})}\left[1\right]$$
$$= 1$$

Substituting back, we obtain

$$\mathbb{P}_{k\sim\mathrm{Unif}(\mathcal{K})}(\Gamma(p,k)=\mathbf{x}) = p(\mathbf{x})\cdot 1 = p(\mathbf{x}).$$

Thus, the watermarked distribution coincides with the original distribution, proving the claim. $\square$

