# OpenReview forum: "MUSE: Model-Agnostic Tabular Watermarking via Multi-Sample Selection"
_ICLR.cc/2026/Conference — ICLR 2026 Poster_

### Official Review · Reviewer_6Ns1 · 2025-10-30

**Soundness:** 2
**Presentation:** 3
**Contribution:** 2
**Rating:** 2
**Confidence:** 5

**Summary:**

The paper proposes a tabular watermark method that assigns a pseudorandom score to each row of the table based on the hash of selected column values and a secret key. During watermark insertion, the model repeatedly samples multiple candidate rows to select the one with the highest score. During detection, the average score of a given table is compared against a threshold to determine whether the table contains the watermark.

**Strengths:**

1. The paper is generally well written and easy to follow.
2. The method 's detectability and fidelity guarantee are supported by mathematical theorems (Theorem 4.1 and Theorem 4.3).
3. The method is also supported by experiments in real world dataset (Adult, Default, Shoppers and Beijing).

**Weaknesses:**

1. The idea seems not novel. https://arxiv.org/abs/2410.02099 and https://arxiv.org/pdf/2403.04808 have almost the same idea as your work though they focus on watermarking large language models.

2. The paper does not consider additive noise attacks in its perturbation experiments. However, such attacks are an important robustness benchmark that has been widely considered in many prior works cited by the authors (https://dl.acm.org/doi/10.1145/3658644.3690373; https://openreview.net/forum?id=71pur4y8gs)

3. It is unclear how to determine a predefined threshold for the detection. From Theorem 4.1, we can see the False Positive Rate control depends on the gap between  the expected score of an unwatermarked sample and the expected score of a watermarked sample obtained via repeated samples. However, in practice this gap is usually unknown since the unwatermarked sample is not necessary from the detector. Also, it does not make sense to consider the "optimal distribution" in Theorem 4.1 since this is in fact the best case. However in practice the unwatermarked table does not always have such nice property.

**Questions:**

NA

---

> ### Author Response · Authors · 2025-11-21
> **Response to Reviewer 6Ns1 (Part I)**
>
> We appreciate your valuable suggestions. We believe your concerns can be addressed through clarification and additional details in our responses below.
>
> ## About novelty
> > The idea seems not novel. https://arxiv.org/abs/2410.02099 and https://arxiv.org/pdf/2403.04808 have almost the same idea as your work though they focus on watermarking large language models.
>
> Thank you for pointing out these two related works. We already discussed WaterMax in our related work section, and we have now added a discussion of Bahri & Wieting (2025), which we were not aware of at submission time. We clarify below why our method is substantially different in **motivation**, **applicability**, and **technical contributions**.
>
> 1. **Motivation for multi-sample selection is fundamentally different in the tabular domain**.
> While WaterMax and Bahri & Wieting (2025) both use multi-sample selection, their motivations are specific to text models:
>     - WaterMax uses repeated sampling because it embeds a watermark without altering the generated text, thereby preserving text quality.
>     - Bahri & Wieting (2025) use it because the procedure remains applicable in black-box LLM settings
>
>    In contrast, our motivation is **unique to tabular generative models**:
>    - **Avoiding inversion of DDIM/diffusion trajectories**.
>      Tabular diffusion models rely on multi-stage normalization, making DDIM inversion error-prone. Repeated sampling avoids inversion entirely and thus avoids both computational cost and inversion-induced error.
>    - **Negligible computational overhead in tabular generation**.
>     Modern tabular generative models (e.g., TabSyn) are lightweight and employ a fast sampler. Hence, multi-sample selection is a practically natural choice.
>
> 2. **Watermarking tabular models is fundamentally different from watermarking autoregressive text models**
>    - **Different generative structure.** LLMs model a **conditional 1D** distribution and generate tokens sequentially, enabling prefix-based hashing in text watermarking. Tabular models generate full rows i.i.d. from **multi-dimensional unconditional distribution**. Since there is no notion of prefix context, text watermarking designs (hashing over a previous window of context) are not applicable. Our work therefore introduces two self-seeded score functions over the full multi-dimensional sample (MUSE-JV and MUSE-PC), a design direction not present in prior text watermarking.
>    - **Different attack model.** Tabular data undergo attacks that do not exist in text watermarking:
>         - row shuffling
>         - row / column / cell deletion
>         - value perturbation (for numerical columns)
>
>      These differ sharply from token-level insertion/deletion/substitution attacks in LLM watermarking. Our design and evaluation explicitly target these table-specific perturbations.
>
>
>
> 3. **Our scoring functions and theory are tailored to the tabular setting**
> Beyond the intuition of repeated sampling, our contributions lie in designing new score functions and analyzing their behavior in the tabular regime:
>    - Our method introduces two score functions (JV and PC hashing) designed to balance distortion, detectability, and robustness in **multi-dimensional i.i.d. row generation**.
>    - Our method provides **theoretical analysis specific to our watermark method**:
>       - characterization of the optimal pseudorandom function for detectability.
>       - derivation of how detectability scales with the number of repeated samples and the number of generated rows, which is fundamentally different from next-token analyses in LLM watermarking.
>       - formulation and proof of a distortion-free property for multi-sample selection in tabular synthesis.

---

> > ### Comment · Reviewer_6Ns1 · 2025-11-21
> >
> > Thank you for the rebuttal.
> >
> > Though the motivation is different, the extension of the idea from https://arxiv.org/abs/2410.02099 to tabular data is relatively straightforward. The idea of section 3 from this earlier paper is quite similar to the idea of section 3.1 in your paper, which is a core part of your paper. Therefore it seems the novelty of your work is quite limited from this perspective.
> >
> > Besides, the fact that this work (https://arxiv.org/abs/2410.02099
> > ) has been put on public for a year makes it difficult to fully accept your explanation that you were unaware of it, especially given the clear overlap on the core idea between this earlier work and your work.
> >
> > Regarding the generative structure, please note that LLMs have recently become increasingly popular for tabular data generation (see https://arxiv.org/abs/2402.17944
> >  for a literature review). Therefore, it seems not convincing to claim novelty from this perspective
> >
> > For the contribution from the perspective of different threat models, I think this is a common issue for all tabular watermarking papers. Therefore, using this point also seems difficult to justify the novelty of the contribution.
> >
> > Besides, for the theoretical contribution you mentioned, the idea of section 4.2 in your paper is similar to Theorem 4.1 in https://arxiv.org/pdf/2410.02099. Therefore the novelty from this perspective is unclear again.

---

> ### Author Response · Authors · 2025-11-21
> **Response to Reviewer 6Ns1 (Part II)**
>
> ## About detection threshold
> >It is unclear how to determine a predefined threshold for the detection. From Theorem 4.1, we can see the False Positive Rate control depends on the gap between the expected score of an unwatermarked sample and the expected score of a watermarked sample obtained via repeated samples. However, in practice, this gap is usually unknown since the unwatermarked sample is not necessary for the detector.
>
> We would like to clarify that the detection threshold is not determined from Theorem 4.1. Instead, it is determined via a standard hypothesis testing procedure: we first fix a desired false positive rate (significance level), and then derive the corresponding threshold based on the distribution of the test statistic under the null hypothesis.
>
> In our work, we use an upper-tail one-sided $z$-test to evaluate the null hypothesis, which is a widely adopted practice in watermark detection (e.g., [1,2,3]). Under the null hypothesis $H_0$ that a table is not watermarked, each row independently receives a score of 0 or 1 with equal probability (i.e., a Bernoulli(0.5) variable). Therefore, the sum of row scores $|W|$ (equivalently, the number of rows with score 1) follows a binomial distribution with mean $N/2$ and variance $N/4$, where $N$ is the number of rows. The corresponding $z$-statistic is:
> $$z=\frac{|W|-N/2}{\sqrt{N/4}} $$
> To determine the detection threshold, we first select a significance level
> $\alpha$ that controls the desired false positive rate (i.e., the probability of incorrectly flagging an unwatermarked table). We then compute the critical value $z_\alpha$ such that
> $$P(Z>z_\alpha|H_0)=\alpha, \ Z\sim \mathcal{N}(0,1)$$
> and declare a table as watermarked if $z>z_\alpha$. Equivalently, this corresponds to a threshold on $|W|$:
> $$
> |W| > \frac{N}{2} + z_\alpha \sqrt{\frac{N}{4}}
> $$
> Finally, the threshold for our detection statistic (Equation 1 in paper) is:
> $$
> S(T):=\frac{|W|}{N} > \frac{1}{2} + z_\alpha \frac{1}{2\sqrt{N}}
> $$
>
> **Difference between $z$-test and Theorem 4.1**
> Although both the upper-tail $z$-test and Theorem 4.1 provide guarantees on the false positive rate, they are fundamentally different in nature. The statistical $z$-test asks: given the null hypothesis that the table is unwatermarked, how likely is it to observe a $z$-statistic at least as large as the one computed? In other words, the $z$-test evaluates the extremeness of the observed statistic **purely under the null distribution**, without relying on any specific properties of the watermarking scheme.
>
> In contrast, Theorem 4.1 characterizes how the false positive rate behaves as a consequence of the **watermarking design itself**. It explicitly captures how the probability of a false positive decreases as the number of repeated samplings $m$ increases and how it depends on the number of watermarked samples $N$. Thus, while the $z$-test gives a generic hypothesis-testing–based guarantee, Theorem 4.1 provides a watermark-specific structural guarantee derived from the mechanics of the embedding process.
>
>
> **References**
>
> [1] Kirchenbauer, John, et al. "A watermark for large language models." International Conference on Machine Learning. PMLR, 2023.
>
> [2] Zhao, Xuandong, et al. "Provable robust watermarking for ai-generated text." International Conference on Learning Representations 2024.
>
> [2] Zhu, Chaoyi, et al. "Tabwak: A watermark for tabular diffusion models." International Conference on Learning Representations 2025.
>
>
> ## About optimal distribution in Thm 4.1
> >Also, it does not make sense to consider the "optimal distribution" in Theorem 4.1 since this is in fact the best case. However in practice the unwatermarked table does not always have such nice property.
>
> We clarify that the "optimal distribution" in Theorem 4.1 refers to the distribution of the watermark score $s_k(x)$, not the distribution of the underlying tabular data $x$. As defined in Equations (2) and (3) (Section 3.2.1), the scoring function is produced by a keyed pseudorandom function (PRF), whose output bit follows certain distribution supported on [0,1]. Therefore, **the distribution of $s_k(x)$ is entirely determined by our watermark design, not by properties of the real data**.
>
> Given this, the role of Theorem 4.1 is to formalize the following question: which score distribution maximizes detectability? Since detectability depends on the separation between watermarked and unwatermarked score statistics, this reduces to minimizing the upper bound in Equation (5). The theorem shows that the Bernoulli(0.5) score distribution uniquely minimizes this bound, and is thus “optimal” in the sense of maximizing the statistical distinguishability required for watermark detection.

---

> > ### Comment · Reviewer_6Ns1 · 2025-11-21
> >
> > Thank you for the rebuttal.
> >
> > For your claim "Under the null hypothesis $H_0$ that a table is not watermarked, each row independently receives a score of 0 or 1 with equal probability (i.e., a Bernoulli(0.5) variable)." This is not true. Imagine a table with only one column, and 30% of the rows take the value of 1, 70% of the rows take the value of 0. Under this circumstance, you can not say that the score they receive from a given hash function is independent from Bernoulli (0.5). The assumption of Bernoulli (0.5) makes sense in papers on watermarking LLMs (For example, https://proceedings.mlr.press/v202/kirchenbauer23a.html) because there are some assumptions of entropy for text, see appendix C from https://openreview.net/forum?id=SsmT8aO45L for a detailed discussion.
> >
> > Since this claim is incorrect for most of the tabular data, I am still very concerned about the reliability of the detection threshold you mentioned.

---

> > > ### Comment · Reviewer_6Ns1 · 2025-11-22
> > >
> > > In summary, most of my concerns are not addressed.
> > >
> > > Therefore I'm sorry that I I cannot give a more encouraging feedback and positive assessment.
> > >
> > > I would like to keep my score of 2 and the confidence of 5. Nonetheless, I encourage you to carefully revise the manuscript and consider submitting it to a future conference, as the topic of tabular data watermarking itself is interesting.

---

> > ### Author Response · Authors · 2025-11-22
> > **Response to Reviewer 6Ns1 (Part III)**
> >
> > ## About Gaussian additive noise attack
> > >The paper does not consider additive noise attacks in its perturbation experiments. However, such attacks are an important robustness benchmark that has been widely considered in many prior works cited by the authors (https://dl.acm.org/doi/10.1145/3658644.3690373; https://openreview.net/forum?id=71pur4y8gs)
> >
> > 1. **Why we did not include Gaussian additive noise**
> > Our initial submission did not include the additive Gaussian noise experiment because we strictly followed the official TabWak codebase when reproducing all five perturbation attacks. Although the TabWak paper describes its Gaussian noise attack as additive (noise std proportional to the cell value), the released implementation （https://github.com/chaoyitud/TabWak/blob/74f1eceb490e95a468aad4db7a661fbd60c6d2b7/watermark/detection.py#L272） uses a multiplicative alteration (although named as 'noise') instead of additive noise, therefore, we rename this attack as 'alteration' in the paper.
> >
> > 2. **Additional experiment: real Gaussian additive noise attack**
> > To fully address the reviewer’s concern, we additionally implement the true Gaussian additive noise attack defined as follows:
> > $$x'=x+\mathcal{N}(0,(\alpha|x|)^2)$$
> > where the attack strength $\alpha$ specifies the standard deviation relative to the original cell value. We report **TPR@1%FPR** across six attack strengths $[0,0.2,0.4,0.6,0.8,1.0]$ on four datasets, with $N=500,m=2$ (using the exact same experimental setup as the other perturbation attacks described in Sec. 5.3).
> >
> >     **Adult**
> >
> >     | Method   | $\alpha$=0    | $\alpha$=0.2  | $\alpha$=0.4 | $\alpha$=0.6 | $\alpha$=0.8 | $\alpha$=1    |
> >     |----------|------|------|-----|-----|-----|------|
> >     | TabWak   | 0.89 | 0.91 | 0.70 | 0.67 | 0.47 | 0.27 |
> >     | TabWak*  | 1.00 | 0.91 | 1.00 | 1.00 | 1.00 | 1.00 |
> >     | MUSE-JV  | 1.00 | 1.00 | 0.39 | 0.18 | 0.11 | 0.03 |
> >     | MUSE-PC  | **1.00** | **1.00** | **1.00** | **1.00** | **1.00** | **1.00** |
> >
> >
> >     **Default**
> >
> >     | Method   | $\alpha$=0    | $\alpha$=0.2  | $\alpha$=0.4 | $\alpha$=0.6 | $\alpha$=0.8 | $\alpha$=1    |
> >     |----------|------|------|-----|-----|-----|------|
> >     | TabWak   | 0.98 | 0.76 | 0.84 | 0.49 | 0.42 | 0.38 |
> >     | TabWak*  | 1.00 | 0.97 | 0.96 | 0.77 | 0.38 | 0.82 |
> >     | MUSE-JV  | 1.00 | 0.84 | 0.68 | 0.62 | 0.54 | 0.61 |
> >     | MUSE-PC  | **1.00** | **1.00** | **1.00** | **1.00** | **1.00** | **1.00** |
> >
> >     **Shoppers**
> >
> >     | Method   | $\alpha$=0    | $\alpha$=0.2  | $\alpha$=0.4 | $\alpha$=0.6 | $\alpha$=0.8 | $\alpha$=1    |
> >     |----------|------|------|-----|-----|-----|------|
> >     | TabWak   | 0.69 | 0.58 | 0.73 | 0.82 | 0.53 | 0.14 |
> >     | TabWak*  | 0.83 | 1.00 | 1.00 | 1.00 | 0.67 | 0.24 |
> >     | MUSE-JV  | 1.00 | 0.73 | 0.48 | 0.49 | 0.02 | 0.10 |
> >     | MUSE-PC  | **1.00** | **1.00** | **1.00** | **1.00** | **1.00** | **1.00** |
> >
> >     **Beijing**
> >
> >     | Method   | $\alpha$=0    | $\alpha$=0.2  | $\alpha$=0.4 | $\alpha$=0.6 | $\alpha$=0.8 | $\alpha$=1    |
> >     |----------|------|------|-----|-----|-----|------|
> >     | TabWak   | 0.81 | 1.00 | 1.00 | 1.00 | 1.00 | 1.00 |
> >     | TabWak*  | 1.00 | 1.00 | 1.00 | 1.00 | 1.00 | 1.00 |
> >     | MUSE-JV  | 1.00 | 0.00 | 0.00 | 0.00 | 0.00 | 0.00 |
> >     | MUSE-PC  | **1.00** | **1.00** | **1.00** | **1.00** | **1.00** | **1.00** |
> >
> >     From the experimental results above, we can oberve that **MUSE-PC consitently achieves the best robustness aginst additive gaussian noise, maintaining 1.0 TPR@1%FPR acroos all datasets and attack strengths**. MUSE-JV exhibits dataset-dependent robustness, reflecting its design goal of minimizing distortion rather than maximizing robustness, consistent with our tradeoff analysis.
> >
> >
> > **Update to the manuscript**
> > We have added the result on Gaussian perturbation attack in Sec 5.3.

---

> > > ### Comment · Reviewer_6Ns1 · 2025-11-22
> > >
> > > Thank you for the rebuttal.
> > >
> > > In the updated experiments in your paper, I notice that the noise is added to the continuous columns. My impression is that MUSE-PC performs well under this circumstance because the categorical columns were not perturbed in this experiment.
> > >
> > > Could you also consider a dataset in which all columns are numerical (i.e., no categorical features)? In that case, would MUSE-PC still be better than TabWak?

---

> ### Author Response · Authors · 2025-11-22
> **Response to Reviewer 6Ns1 (Part IV)**
>
> Thank you for your quick reply. We appreciate the opportunity to further clarify.
>
> **R1**
> > Though the motivation is different, the extension of the idea from https://arxiv.org/abs/2410.02099 to tabular data is relatively straightforward.
>
> We respectfully disagree that extending Bahri & Wieting (2025) or WaterMax to tabular data is straightforward. As we have outlined in previous rebuttals, there are fundamental differences in both generative structure and attack surfaces between textual and tabular domains:
>
> Tabular watermarking handles i.i.d samples of a multi-dimensional distribution (corresponds to i.i.d rows and multiple columns), and must be robust to unique attack types.
>
> Our method proposed two methods (JC and PC hashing) to explicitly handle the above scenario and carefully designed to tradeoff distortion, detectability and robustness (please see Section 3.2 in our paper for a complete discussion). In contrast, the specific data structure of tabular data renders n-gram LLM watermarking schemes vulnerable to row shuffling attack, **even when using an autoregressive tabular generative model**. We invite the reviewer to examine the concrete example provided in **R4**.
>
> We hope this concrete example clearly demonstrates that extending LLM watermarking techniques to the tabular domain is far from trivial.
>
> **R2**
> > The idea of section 3 from this earlier paper is quite similar to the idea of section 3.1 in your paper, which is a core part of your paper. Therefore, it seems the novelty of your work is quite limited from this perspective.
>
> We appreciate the opportunity to clarify the structure of our contribution. We respectfully point out that Section 3.1 is not intended to represent the core technical novelty of our work. Rather, its purpose is to formalise the general generation–watermarking–detection pipeline to establish the problem setting.
>
> The core methodological novelty lies in Section 3.2, where we introduce two specific hashing schemes—JV and PC hashing—that constitute the heart of our proposal. Unlike the general framework in Section 3.1, these schemes are specifically tailored to the discrete and heterogeneous nature of tabular data. They are novel because:
>
> - They effectively leverage the generative structure of tabular models to balance detectability and distributional fidelity.
> - They are explicitly designed to provide robustness against attack vectors unique to the tabular domain.
>
>
> **R3**
> > Besides, the fact that this work (https://arxiv.org/abs/2410.02099 ) has been put on public for a year makes it difficult to fully accept your explanation that you were unaware of it, especially given the clear overlap on the core idea between this earlier work and your work.
>
> We agree that the missing citation of Bahri & Wieting (2025) is an oversight on our part, and we have now added a discussion of this work in the revised manuscript.
>
> However, **we believe the omission does not affect the substance of our contributions nor our positioning within prior work**.
>
> - Our original submission already cited and discussed two closely related works [1,2] that employ the same high-level paradigm of watermarking generative models via repeated sampling. This demonstrates that **we were not attempting to sidestep this general line of work**, but rather that we unfortunately missed one specific paper in a rapidly growing literature.
> - To the best of our knowledge, **Bahri & Wieting 2025 (https://arxiv.org/abs/2410.02099) is currently available only as an Arxiv preprint (posted Oct 2024) and has not yet been published in peer-reviewed conference proceedings nor journals**. Consequently, it falls under the ICLR policy regarding preprints (https://iclr.cc/Conferences/2026/ReviewerGuide), which states:
>     > Note that arXiv is not considered a peer-reviewed venue. As such, authors are not required to compare to papers solely on arXiv: they may be excused for not knowing about papers not published in peer-reviewed conference proceedings or journals, which includes papers exclusively available on arXiv.
>
>     > Reviewers can make authors aware of related contemporaneous work or arXiv papers, but the lack of such comparisons cannot be a basis for rejection.
>
> Finally, we deeply respect this work and we thank the reviewer for making us aware of this related work.
>
> **References**
>
> [1] Dathathri, S., See, A., Ghaisas, S. et al. Scalable watermarking for identifying large language model outputs. Nature 634, 818–823 (2024)
>
> [2] Giboulot, Eva, and Teddy Furon. "WaterMax: breaking the LLM watermark detectability-robustness-quality trade-off." Advances in Neural Information Processing Systems 37 (2024): 18848-18881.

---

> ### Author Response · Authors · 2025-11-22
> **Response to Reviewer 6Ns1 (Part V)**
>
> **R4**
> >Regarding the generative structure, please note that LLMs have recently become increasingly popular for tabular data generation (see https://arxiv.org/abs/2402.17944 for a literature review). Therefore, it seems not convincing to claim novelty from this perspective
>
> We appreciate the reviewer's pointer to LLM-based tabular generative models (and our experiment actually included an autoregressive model, [1] in Section 5.4). Below, we will detail why our work has novelty in handling the specific generative structure of tables.
>
> **First**. We claim that **even when the underlying model class is autoregressive, the standard prefix-based text watermarking paradigm cannot be directly applied to tables**:
>
> - In text generation, **tokens are causally correlated**; standard watermarks exploit this property by using the preceding context to seed the random number generator for the next token.
> - In table generation, **tabular data consists of i.i.d. rows and is generated (and later stored/used) as an unordered set of rows.**.
>
> This difference in generative structure has a crucial implication for watermarking: **any scheme that ties watermark randomness to a 'prefix' is brittle for tabular watermarking, as a simple row shuffling would destroy the watermark**. Thus, even if one parametrizes a tabular generator autoregressively, directly reusing prefix-based text watermarking is inappropriate for the tabular setting. A related observation has also been made in TabWak [2] when contrasting tabular watermarking with image watermarking, where the i.i.d nature of table synthesis is described as the lack of a "diversity enhancer."
>
> **Second**, since different types of generative models (e.g., diffusion[3], AR[1], masked model[4]) exist for table synthesis, it is therefore desirable for a watermarking scheme to be model-agnostic. Our method achieves this via multi-sample selection, which explores a direction that is not covered by existing text watermarking methods.
>
> **References**
>
> [1] Castellon, Rodrigo, et al. "Dp-tbart: A transformer-based autoregressive model for differentially private tabular data generation." arXiv preprint arXiv:2307.10430 (2023).
>
> [2] Zhu, Chaoyi, et al. "Tabwak: A watermark for tabular diffusion models." International Conference on Learning Representations. ICLR 2025.
>
> [3] Zhang, Hengrui, et al. "Mixed-type tabular data synthesis with score-based diffusion in latent space." International Conference on Learning Representations. ICLR 2024.
>
> [4] Zhang, Hengrui, et al. "TabNAT: A Continuous-Discrete Joint Generative Framework for Tabular Data." Forty-second International Conference on Machine Learning.
>
>
> **R5**
> > For the contribution from the perspective of different threat models, I think this is a common issue for all tabular watermarking papers. Therefore, using this point also seems difficult to justify the novelty of the contribution.
>
> The threat models and attacks we consider (e.g., row/column deletion, cell perturbation) are indeed common in the tabular watermarking literature, as we follow the common setting to conduct a fair comparison with previous works. **We do not claim novelty in introducing these attacks**.
>
> **Our intention was instead to highlight how these table-specific attacks differ from typical text watermarking attacks**. This was directly in response to the reviewer’s concern that our work offers limited novelty beyond text watermarking methods such as Bahri & Wieting (2025) (W1 in the original review).

---

> ### Author Response · Authors · 2025-11-22
> **Response to Reviewer 6Ns1 (Part VI)**
>
> **R6**
> > Besides, for the theoretical contribution you mentioned, the idea of section 4.2 in your paper is similar to Theorem 4.1 in https://arxiv.org/pdf/2410.02099. Therefore, the novelty from this perspective is unclear again.
>
> We thank the reviewer for highlighting the connection to Bahri & Wieting (2025). Below, we clarify that our theoretical contributions consist of two conceptually distinct parts (detactability and distortion-free). Although only the 'distortion-free' analysis is mentioned by the reviewer, we aim to provide a full picture of the novelty of our contribution on the theoretical side. We further show that the core novelty of our theoretical results is not captured by the comparison to Bahri & Wieting (2025).
>
> 1. **Section 4.1 – Detectability Analysis**
>   This part contains two contributions that **do not** appear in Bahri & Wieting (2025), WaterMax, or any prior watermarking paper (to our knowledge):
>     - **We characterize how properties of the pseudorandom function (PRF) influence watermark detectability**.
>     This question has not been studied in the watermarking literature. Our analysis shows which classes of PRFs are provably more favorable for detection.
>     - **We derive how detectability scales with the number of sampling repetitions $m$ and number of rows $N$**.
>     This is also unique to our method and the tabular watermark setting.
>
>    Thus, the detectability analysis in Section 4.1 is new, specific to our watermark construction, and fundamentally different from the theory in Bahri & Wieting (2025).
>
> 2. **Section 4.2 – Distortion-free Analysis: definition and assumption**
> Our contribution in Section 4.2 consists of two parts:
>    - **About definition**
>    We are the first to formally define distortion-freeness for tabular generative watermarking: the existing generative watermarking method for tabular data, TabWak [1] did not provide a theoretical analysis on distortion-freeness. And post-processing tabular watermarking methods quantify distortion via some distance metric between watermarked and unwatermarked table [2,3]. Therefore, we believe our definition provides a novel and useful concept to the tabular watermarking domain.
>
>       Now we compare the definition of distortion-free with Bahri & Wieting (2025):
>       - Bahri & Wieting (2025):
>       Distortion-free is defined as for any sequence of tokens, the probability of generating that sequence is identical for watermarked and unwatermarked LLMs.
>       - Our definition (Definition 4.2)
>       Distortion-free is defined as for any sequence of rows, the joint distribution produced by the watermarked generator equals the **product of the marginal distributions** of the unwatermarked generator.
>
>      The key difference is that our definition is tailored to i.i.d. row-wise generative processes, which is essential for tabular data. Thus, the theoretical formulation itself is novel and domain-specific.
>
>    - **About assumption**
>      - Bahri & Wieting (2025): the condition for distortion-free to hold is that unique n-grams across the sampled text sequences are **independent**.
>      - Our work:  we assume the selected column values for seeding the random number generator are **unique** across i.i.d. sampled rows, and the number of repeated samples is 2.
>
>      We acknowledge that while the two assumptions are presented in different forms, both Bahri & Wieting (2025) and our work rely on assumptions that guarantee independence when moving from per-token/per-row distortion-free to sequence-level distortion-free. However, we believe the high-level reasoning structure is similar due to the shared framework of multi-sample selection; thus, this does not diminish the novelty of our work.
>
>
> **References**
>
> [1] Zhu, Chaoyi, et al. "Tabwak: A watermark for tabular diffusion models." International Conference on Learning Representations. ICLR 2025.
>
> [2] He, Hengzhi, et al. "Watermarking generative tabular data." arXiv preprint arXiv:2405.14018 (2024).
>
> [3] Gu, Bochao, Hengzhi He, and Guang Cheng. "Watermarking Generative Categorical Data." arXiv preprint arXiv:2411.10898 (2024).

---

> > ### Comment · Reviewer_6Ns1 · 2025-11-23
> >
> > For R6, thank you for your clarification.
> >
> > You state that both your work and previous work “rely on assumptions that guarantee independence when moving from per-token/per-row distortion-free to sequence-level distortion-free.” and that “the high-level reasoning structure is similar due to the shared framework of multi-sample selection”.
> >
> > Since the two methods indeed have a shared framework and rely on the same type of assumptions, then this seems to suggest that the theoretical result is not fundamentally new. In other words, this seems not be able to support novelty on the theoretical side.

---

> ### Author Response · Authors · 2025-11-22
> **Response to Reviewer 6Ns1 (Part VII)**
>
> **R7**
> > Your claim "Under the null hypothesis that a table is not watermarked, each row independently receives a score of 0 or 1 with equal probability (i.e., a Bernoulli(0.5) variable)." is not true.
>
> > The assumption of Bernoulli (0.5) makes sense in papers on watermarking LLMs because there are some assumptions of entropy for text.
>
> We thank the reviewer for the insightful comment.
>
> 1. We agree that when a column has extremely low entropy (e.g., binary), directly hashing that column may not yield independent Bernoulli(0.5) output due to row collisions. However, we note that **this low-entropy problem is precisely a challenge we explicitly address in JV-hashing design**. Specifically, this is why our JV-hashing design hashes the entire vector of selected values as a concatenated vector. By doing this, we essentially boost the entropy of the input to the PRF. As long as the combined tuples differ across rows, the hash outputs act as independent random variables. (For extreme cases where the entire dataset has extremely low entropy, we discuss the limitations in the "Complexity of Mixed-type Datasets" section in our response to Reviewer **UNgJ**.)
>
> 2. Modeling the output of a keyed pseudorandom function follows a certain distribution is a standard assumption in watermarking literature to derive theoretical bounds and hypothesis tests [1, 2, 3].
>
> 3. Empirically, we observe our hashing designs do allocate enough entropy s.t. that its output bit follows a desired distribution, e.g., Bernoulli (0.5). As shown in Figure 10, the distribution of the output bits closely aligns with Bernoulli(0.5) in our experiments. This confirms that, in practice, the assumption is valid for the datasets tested.
>
> 4. We note that another way to address this issue is to estimate the statistic under the null hypothesis via Monte Carlo simulation, as in TabWak[4]. However, we note that characterizing the null distribution by either theoretical assumption (our approach) or by Monte Carlo estimation (TabWak) is primarily a methodological choice for formulating the hypothesis test. The exact shape of the theoretical null distribution does not dictate the system's inherent detectability. Indeed, our core evaluation relies on AUC-ROC and TPR@0.1%FPR, which directly quantify the empirical separation between the watermarked and unwatermarked populations, irrespective of the theoretical derivation.
>
> **References**
>
> [1] Kirchenbauer, John, et al. "A watermark for large language models." International Conference on Machine Learning. PMLR, 2023.
>
> [2] Zhao, Xuandong, et al. "Provable robust watermarking for ai-generated text." ICLR 2024.
>
> [3] Giboulot, Eva, and Teddy Furon. "WaterMax: breaking the LLM watermark detectability-robustness-quality trade-off." Advances in Neural Information Processing Systems 37 (2024): 18848-18881.
>
> [4] Zhu, Chaoyi, et al. "Tabwak: A watermark for tabular diffusion models." International Conference on Learning Representations. ICLR 2025.

---

> ### Comment · Reviewer_6Ns1 · 2025-11-22
>
> Thank you for the rebuttal.
>
> For R1 and R4, regarding your claim that "any scheme that ties watermark randomness to a 'prefix' is brittle for tabular watermarking, as a simple row shuffling would destroy the watermark", please note that as early as 2023, the work https://openreview.net/pdf?id=Bwz0fy9Hc9 proposed a watermark using a fixed Green-Red split that does not rely on prefixes, which is therefore robust to shuffling. Therefore it seems not correct to say that row shuffling is an additional challenge for extending watermarking methods for LLM to tabular watermark.

---

> ### Comment · Reviewer_6Ns1 · 2025-11-23
>
> For R2 and R3, you wrote: “We present MUSE as a general paradigm
> and introduce two specific implementations that navigate the crucial trade-off between data fidelity
> and watermark detectability/robustness: (1) Joint-Vector (JV) hashing, tailored for minimal distortion
> (distribution-preserving), and (2) Per-Column (PC) hashing, designed for maximal robustness and
> detectability. ” in the paper.
> From this sentence, my understanding is that Section 3.1 corresponds to the general paradigm you propose. However, this part is quite similar to what has been discussed in earlier work (section 3 in the **published Watermax paper** and also section 3 in https://arxiv.org/abs/2410.02099). Therefore much of your contribution seems to fall within this more general framework, with JV and PC mainly representing engineering-style incremental variants. Therefore in my opinion, the level of novelty may not be strong enough for a top conference like ICLR.
>
> **To be clear, my assessment is not based on missing citations.** Rather, even only compared with the already published Watermax work, I feel that the conceptual novelty is limited, and this is one of the main reasons for my score.

---

> ### Comment · Reviewer_6Ns1 · 2025-11-23
>
> For R5,  for your claim "Our intention was instead to highlight how these table-specific attacks differ from typical text watermarking attacks", it seems many of them have analogues in the field of large language models. For example, cell perturbation corresponds to token replacement, and row shuffling corresponds to changing the order of sentences or phrases. Because of these parallels, I do not think this difference meaningfully supports the novelty argument.

---

> ### Comment · Reviewer_6Ns1 · 2025-11-23
>
> For R7, you claim "We agree that when a column has extremely low entropy (e.g., binary), directly hashing that column may not yield independent Bernoulli(0.5) output due to row collisions. However, we note that this low-entropy problem is precisely a challenge we explicitly address in JV-hashing design. Specifically, this is why our JV-hashing design hashes the entire vector of selected values as a concatenated vector. By doing this, we essentially boost the entropy of the input to the PRF. As long as the combined tuples differ across rows, the hash outputs act as independent random variables. (For extreme cases where the entire dataset has extremely low entropy, we discuss the limitations in the "Complexity of Mixed-type Datasets" section in our response to Reviewer UNgJ.)"
>
> However, JV-hashing seems not be able to perform well when facing additive-noise attacks, as shown in your own experiments in the rebuttal. Therefore I'm still concerned about using it in practice, so I respectfully disagree this offers a satisfactory solution.
>
> For your claim "Modeling the output of a keyed pseudorandom function follows a certain distribution is a standard assumption in watermarking literature to derive theoretical bounds and hypothesis tests [1, 2, 3].", as I explain before, this is common in literature in LLM watermarking because the text usually has high entropy. However this is not the case for tabular data.
>
> For your claim "Empirically, we observe our hashing designs do allocate enough entropy s.t. that its output bit follows a desired distribution, e.g., Bernoulli (0.5). As shown in Figure 10, the distribution of the output bits closely aligns with Bernoulli(0.5) in our experiments. This confirms that, in practice, the assumption is valid for the datasets tested.", this is valid for the dataset you choose in the paper, but not for all common datasets. As I explain before, let us consider a table with only one column, and 30% of the rows take the value of 1, 70% of the rows take the value of 0. This assumption fails in this dataset.
>
> For your claim "We note that another way to address this issue is to estimate the statistic under the null hypothesis via Monte Carlo simulation, as in TabWak[4]. However, we note that characterizing the null distribution by either theoretical assumption (our approach) or by Monte Carlo estimation (TabWak) is primarily a methodological choice for formulating the hypothesis test. The exact shape of the theoretical null distribution does not dictate the system's inherent detectability. Indeed, our core evaluation relies on AUC-ROC and TPR@0.1%FPR, which directly quantify the empirical separation between the watermarked and unwatermarked populations, irrespective of the theoretical derivation." This is true. Please discuss this point more carefully in the paper to give a clearer explanation on the limitations of your approach in detection, and this potential way to mitigate this issue.

---

> > ### Comment · Reviewer_6Ns1 · 2025-11-23
> >
> > Thank you again for your rebuttal. However, as I explained in detail, most of my concerns are still not addressed. Therefore, I would like to keep my original score of 2 and confidence of 5.

---

> ### Author Response · Authors · 2025-11-25
> **Response to Reviewer 6Ns1 (Part VIII)**
>
> We greatly appreciate the time and effort you spent reviewing our work, and we believe the suggestions will significantly improve the quality of our paper.
>
> **R7**
> > Could you also consider a dataset in which all columns are numerical (i.e., no categorical features)? In that case, would MUSE-PC still be better than TabWak?
>
> We appreciate your insightful question.
>
> **TL;DR** MUSE-PC, equipped with a quantisation step before score computing, consistently outperform TabWak/TabWak* on both data quality and detectability/robustness.
>
> In response to your question, we have added two new datasets, **California** and **Letter**, with only numerical columns.(see dataset details in Appendix B.2 in [1]).
>
> As a reminder, the original MUSE-PC design in the paper primarily targets a **threat model where the adversary perturbs a subset of values with arbitrary strength, but some values in each row remain unchanged**. The robustness of MUSE-PC in this regime comes from spreading the watermark signal across all columns: each column contributes independently to the per-row score, so as long as some values are preserved, the aggregated signal remains detectable.
>
> However, in the setting where **every entry** is perturbed by small Gaussian noise, directly computing the score on raw features becomes more sensitive than desired.
>
> **To handle this case, we introduce a simple normalisation step before computing the MUSE-PC score**.
>
> Concretely, we apply a transformation $f$ such that $f(x) \approx f(x+z)$ when $z$ is a small perturbation, so that the downstream score is stable even if all entries receive small noise. In practice, we instantiate $f$ as a **quantisation in the log domain**: we first map each numerical value to its log scale and then assign it to one of a fixed number of bins (this introduces the number of bins as a hyperparameter). The log transform makes the bin widths adaptive (larger $|x|$ receives a wider bin), which matches the intuition that larger-magnitude values can tolerate larger absolute perturbations. This preprocessing is lightweight and does not change the procedure of MUSE-PC.
>
> We set the  number of bins as 32 and compare the detectability, distortion and robustness of TabWak and MUSE-PC:
>
> ### Detectability vs Distortion
> **Letter**
> |  | **Marg.** | **Corr.** | **C2ST** | **MLE** | **T@F** | **AUC** |
> | :--- | :--- | :--- | :--- | :--- | :--- | :--- |
> | no-wm | 0.975 | 0.98 | 0.98 | 0.992 | - | - |
> | TabWak | **0.928** | 0.938 | 0.685 | 0.926 | 0.9 | 0.9985 |
> | TabWak* | 0.922 | 0.93 | 0.607 | 0.919 | **1** | **1** |
> | MUSE-PC | **0.928** | **0.964** | **0.74** | **0.99** | **1** | **1** |
>
> **California**
> |  | **Marg.** | **Corr.** | **C2ST** | **MLE** | **T@F** | **AUC** |
> | :--- | :--- | :--- | :--- | :--- | :--- | :--- |
> | no-wm | 0.992 | 0.992 | 0.995 | 0.994 | - | - |
> | TabWak | 0.905 | 0.937 | 0.783 | 0.787 | 0.39 | 0.8707 |
> | TabWak* | 0.891 | 0.93 | 0.753 | 0.934 | 0.53 | 0.9755 |
> | MUSE-PC | **0.933** | **0.964** | **0.851** | **0.994** | **1** | **1** |
>
> ### Robustness to Gaussian perturbation
> **Letter**
> |  | $\alpha=0$ | $\alpha=0.2$ | $\alpha=0.4$ | $\alpha=0.6$ | $\alpha=0.8$ | $\alpha=1$ |
> | :--- | :--- | :--- | :--- | :--- | :--- | :--- |
> | TabWak | 0.87 | 0.93 | 0.82 | 0.62 | 0.39 | 0.43 |
> | TabWak* | 0.98 | 1 | 0.99 | 0.95 | 0.9 | 0.9 |
> | MUSE-PC | **1** | **1** | **1** | **1** | **1** | **0.995** |
>
>
> **California**
> |  | $\alpha=0$ | $\alpha=0.2$ | $\alpha=0.4$ | $\alpha=0.6$ | $\alpha=0.8$ | $\alpha=1$ |
> | :--- | :--- | :--- | :--- | :--- | :--- | :--- |
> | TabWak | 0.39 | 0.24 | 0.07 | 0 | 0.06 | 0.15 |
> | TabWak* | 0.53 | 0.23 | 0.23 | 0.15 | 0.19 | 0.22 |
> | MUSE-PC | **1** | **1** | **0.855** | **0.79** | **0.485** | **0.345** |
>
> **Observations:**
> - **MUSE-PC consistently outperforms TabWak and TabWak\* on both distortion and robustness, across two datasets**.
> - TabWak exhibits good robustness on the Letter dataset but performs very poorly on California. We hypothesize that this discrepancy arises from the instability involved in reversing the entire sampling pipeline.
>
> Next, we conduct an ablation study on the number of quantisation bins.
> ### Ablation study for the number of bins
> We evaluate how data quality and robustness of MUSE-PC change across a range of bin numbers (16, 32, 64, 128, 256)
>
> **How data quality change w.r.t num_bin?**
> **Letter**
> | **bin_num** | **Marg.** | **Corr.** | **C2ST** | **MLE** |
> | :--- | :--- | :--- | :--- | :--- |
> | 16 | 0.93 | 0.961 | 0.778 | 0.99 |
> | 32 | 0.928 | 0.964 | 0.74 | 0.99 |
> | 64 | 0.94 | 0.965 | 0.836 | 0.993 |
> | 128 | 0.935 | 0.966 | 0.824 | 0.99 |
> | 256 | 0.942 | 0.965 | 0.836 | 0.991 |
>
> **California**
> | **bin_num** | **Marg.** | **Corr.** | **C2ST** | **MLE** |
> | :--- | :--- | :--- | :--- | :--- |
> | 16 | 0.923 | 0.968 | 0.838 | 0.993 |
> | 32 | 0.933 | 0.964 | 0.851 | 0.994 |
> | 64 | 0.936 | 0.964 | 0.848 | 0.994 |
> | 128 | 0.949 | 0.968 | 0.864 | 0.994 |
> | 256 | 0.958 | 0.973 | 0.874 | 0.993 |

---

> ### Author Response · Authors · 2025-11-25
> **Response to Reviewer 6Ns1 (Part Ⅸ)**
>
> continue R7...
>
> The following figure visualises the trend that data quality (average of Marg., Corr., C2ST and MLE) increase as num_bin increases.
>
> [Figure A. Ablation analysis: Data quality vs. num of bins](https://i.postimg.cc/4423BDkc/distortion-num-bin.png)
>
>
> **How robustness change w.r.t num_bin?**
>
> **Letter**
> | **bin_num** | $\alpha=0$ | $\alpha=0.2$ | $\alpha=0.4$ | $\alpha=0.6$ | $\alpha=0.8$ | $\alpha=1.0$ |
> | :--- | :--- | :--- | :--- | :--- | :--- | :--- |
> | 16 | 1 | 1 | 1 | 1 | 1 | 1 |
> | 32 | 1 | 1 | 1 | 1 | 1 | 0.995 |
> | 64 | 1 | 1 | 0.78 | 0.47 | 0.585 | 0.326 |
> | 128 | 1 | 0.97 | 0.485 | 0.25 | 0.175 | 0.21 |
> | 256 | 1 | 0.725 | 0.495 | 0.43 | 0.475 | 0.33 |
>
> **California**
> | **bin_num** | $\alpha=0$ | $\alpha=0.2$ | $\alpha=0.4$ | $\alpha=0.6$ | $\alpha=0.8$ | $\alpha=1.0$ |
> | :--- | :--- | :--- | :--- | :--- | :--- | :--- |
> | 16 | 1 | 1 | 0.995 | 0.935 | 0.685 | 0.65 |
> | 32 | 1 | 1 | 0.855 | 0.79 | 0.485 | 0.345 |
> | 64 | 1 | 0.885 | 0.73 | 0.475 | 0.455 | 0.525 |
> | 128 | 1 | 0.685 | 0.29 | 0.285 | 0.555 | 0.465 |
> | 256 | 1 | 0.35 | 0.315 | 0.15 | 0.29 | 0.185 |
>
>
> The following figure visualises the trend that robustness decreases as num_bin increases.
>
> [Figure B. Ablation analysis: Robustness vs. num of bins on Letter Dataset](https://i.postimg.cc/VL1c945N/robust-bin-letter.png)
>
> [Figure C. Ablation analysis: Robustness vs. num of bins on California Dataset](https://i.postimg.cc/k4rHJXbK/robust-bin-california.png)
>
> **Observations:**
> - The bin_num introduces a trade-off between distortion and robustness: coarser binning improves robustness but slightly increases distortion, while finer binning improves fidelity at the cost of reduced robustness.
> - Choosing bin_num = 32 is enough to outperform the robustness of TabWak. Even under the coarse quantisation bin_num=16, MUSE-PC still preserve better data quality than TabWak.
>
> **References**
>
> [1] Zhang, Hengrui, et al. "TabNAT: A Continuous-Discrete Joint Generative Framework for Tabular Data." Forty-second International Conference on Machine Learning.
>
> **R8**
> > ...please note that as early as 2023, the work https://openreview.net/pdf?id=Bwz0fy9Hc9 proposed a watermark using a fixed Green-Red split that does not rely on prefixes, which is therefore robust to shuffling. Therefore, it seems not correct to say that row shuffling is an additional challenge for extending watermarking methods for LLM to tabular watermark.
>
> Thank you for your question.
>
> We agree that Unigram [1] does not rely on prefixes and is therefore technically robust to shuffling. However, we clarify below that **this property alone does not make Unigram a viable baseline for tabular watermarking, because directly applying Unigram to tabular generative models introduces severe distributional distortion**, leading to low-fidelity and low-utility synthetic tables.
>
> **Unigram creates substantial distortion when applied to tables**
> In text watermarking, Unigram works by first dividing the vocabulary into a green and red list (say of equal size), then never samples a token from the red list. To apply to tables, let's assume we use an autoregressive tabular generative model and have quantised the table into discrete values. Applying Unigram means **half of the values of each column will never be sampled**, **which will significantly harm the data fidelity and utility**.
>
> **Empirical evidence**
> TabWak [2] explicitly observed that using a fixed control seed across all rows significantly degrades generative quality and utility. This is shown in their Gaussian Shading results, where the fixed-seed constraint significantly limits the "diversity" of the sampling.
>
> **Theoretical evidence**
> Our distortion-free Theorem 4.3 further highlights the limitation: To maintain distortion-free generation under a fixed key, only one row in the table can be safely watermarked without being masked by our *repeated key masking* procedure, which significantly degrades the detectability.
>
> In summary, directly applying unigram will either achieve high distortion or low detectability, resulting in an unfavourable tradeoff.
>
> **Reference**
>
> [1] Zhao, Xuandong, et al. "Provable robust watermarking for ai-generated text." ICLR 2024.
>
> [2] Zhu, Chaoyi, et al. "Tabwak: A watermark for tabular diffusion models." International Conference on Learning Representations. ICLR 2025.

---

> ### Author Response · Authors · 2025-11-25
> **Response to Reviewer 6Ns1 (Ⅹ)**
>
> **R9**
> > ...much of your contribution seems to fall within this more general framework.
>
> We appreciate the reviewer’s perspective, but we emphasise that **falling under a broad categorisation does not preclude novelty in specific methodology**.
>
> > ...with JV and PC mainly representing engineering-style incremental variants.
>
> We wish to clarify that JV and PC hashing designs are not merely engineering-style tweaks, but are fundamental requirements for addressing the unique constraints of the tabular domain. As we have detailed in previous responses **R4** and **R8**, standard hashing methods (either prefix-based or Unigram) used in LLM fail in this context.
>
> **R10**
> >  ... it seems many of them have analogues in the field of large language models. For example, cell perturbation corresponds to token replacement, and row shuffling corresponds to changing the order of sentences or phrases. Because of these parallels, I do not think this difference meaningfully supports the novelty argument.
>
> We acknowledge the conceptual parallels the reviewer draws between tabular and text perturbations. However, **while the mechanics may look similar, the utility constraints imposed by the data structure make the attacks fundamentally different in practice**.
>
> Tabular data possesses unique structural properties that decouple it from text-based intuition:
> - Row Exchangeability: A table is an unordered set of rows (unlike the sequential nature of text).
> - Fixed row length: Every row has the same length and attribute alignment.
> - Heterogeneous Data: Values can be mixed-type (continuous and discrete).
> - Column Permutation Invariance: Changing the order of columns does not alter the meaning of the table.
>
> Crucially, these properties hold regardless of whether the table was generated by an autoregressive model [1,2]. In the following, we list how each property of tabular data induces its unique attack:
> - **Shuffling/permutation (Points 1 & 4)**: Due to exchangeability, an attacker can execute arbitrary row or column shuffling without any loss of semantic meaning or utility. In contrast, reordering sentences, phrases, or words in text requires complex semantic checks to avoid degrading coherence. This is why n-gram-based watermarks work for text (where order is rigid) but would fail in tables (where order is irrelevant).
> - **Column/row/cell Deletion (Point 2)**: Take column deletion as an example, deleting a column corresponds to removing a specific feature across the entire table. The textual analogue is that deleting the token at a fixed position (e.g., the 2nd word) of every sentence would result in syntactic gibberish. Thus, "column deletion" has no functional equivalent in text.
> - **Value Perturbation (Point 3)**: Due to the existence of continuous features in tables, tabular attacks can utilise continuous noise injection or value alteration.
>
> **References**
>
> [1] Castellon, Rodrigo, et al. "Dp-tbart: A transformer-based autoregressive model for differentially private tabular data generation." arXiv preprint arXiv:2307.10430 (2023).
>
> [2] Borisov, Vadim, et al. "Language models are realistic tabular data generators." ICLR 2023.
>
> **R11**
> >You state that both your work and previous work “rely on assumptions that guarantee independence when moving from per-token/per-row distortion-free to sequence-level distortion-free,” and that “the high-level reasoning structure is similar due to the shared framework of multi-sample selection”.
>
> > Since the two methods indeed have a shared framework and rely on the same type of assumptions, then this seems to suggest that the theoretical result is not fundamentally new. In other words, this seems not be able to support novelty on the theoretical side.
>
> We respectfully disagree with the assessment that our theoretical results lack novelty due a similar technical assumption with prior work. **Conflating this single similar assumption with the entirety of our theoretical contribution overlooks the other parts of our theoretical analysis that are fundamentally new and distinct from existing literature**:
>
> 1. **Section 4.1 – Detectability analysis**.
> As detailed in our previous response (**R6**), Sec. 4.1 studies detectability, and contains two results that, to the best of our knowledge, do not appear in Bahri & Wieting (2025), WaterMax, or any prior watermarking paper.
>
> 2. **Section 4.2 – Distortion-free analysis: new definition and assumption**.
> As detailed in our previous response (**R6**), our contribution is also two parts:
>     - **New definition**: We provide the first formal definition of "distortion-free" for tabular generative models (Definition 4.2).
>     - **Similar technical assumptions and reasoning structure**: as we have detailed in **R6**, we believe the high-level reasoning structure is similar due to the shared framework of multi-sample selection, which does not diminish the novelty of our work.

---

> > ### Comment · Reviewer_6Ns1 · 2025-11-25
> >
> > Thank you for the rebuttal.
> >
> > For R9 and R10, I believe the responses are also satisfactory.
> >
> > For R11, I still keep my original opinion on the novelty of this theoretical component, since as the authors acknowledge themselves, "high-level reasoning structure is similar due to the shared framework of multi-sample selection".

---

> ### Author Response · Authors · 2025-11-25
> **Response to Reviewer 6Ns1 (ⅩI)**
>
> **R12**
> > JV-hashing seems not be able to perform well when facing additive-noise attacks, as shown in your own experiments in the rebuttal. Therefore I'm still concerned about using it in practice, so I respectfully disagree this offers a satisfactory solution.
>
> We clarify that JV-hashing's vulnerability to perturbation attack is expected, as explained in section 3.2.2 in the paper; the design’s “all-or-nothing” nature makes it fragile: any modification to a selected value invalidates the entire watermark.
> We note that PC-hashing, as expected to overcome this issue, consistently achieves the best robustness to the attack. JV and PC hashing represent two points on the tradeoff between distortion and detectability/robustness. If security is what the user cares more about, PC hashing will be preferred.
>
>
> **R12**
> > For your claim "Modeling the output of a keyed pseudorandom function follows a certain distribution is a standard assumption in watermarking literature to derive theoretical bounds and hypothesis tests [1, 2, 3].", as I explain before, this is common in literature in LLM watermarking because the text usually has high entropy. However, this is not the case for tabular data.
>
> We respectfully disagree with the implication that “tabular data inherently has lower entropy than text.” **Whether the input to a keyed PRF has enough entropy is fundamentally a dataset-level property rather than a modality-level one**. Even within the text modality, entropy varies widely across models and tasks—for example, post-training procedures such as RLHF are known to substantially reduce output entropy in practice [1]. Thus, it is not accurate to claim that text universally has high entropy while tables universally do not.
>
> Therefore, instead of making modality-level claims, we take a **pragmatic approach** and empirically validate the effectiveness of our watermark. As our experiments show, for the realistic tabular datasets we consider, the effective entropy (especially after our hashing/aggregation design) is sufficient to support reliable watermark embedding and detection.
>
>
> > ...This is true. Please discuss this point more carefully in the paper to give a clearer explanation on the limitations of your approach in detection, and this potential way to mitigate this issue.
>
> Thanks for your valuable suggestion. We will add a section to thoroughly discuss using Monte Carlo estimation to construct the hypothesis test.
>
>
> **Reference**
>
> [1] Kirk, Robert, et al. "Understanding the effects of rlhf on llm generalisation and diversity." ICLR 2024.

---

> > ### Comment · Reviewer_6Ns1 · 2025-11-25
> >
> > For R12, I think the response is also satisfactory now.

---

> > ### Comment · Reviewer_6Ns1 · 2025-11-25
> >
> > In summary, although there is a slight difference of opinion regarding the novelty on some theorems, which is subjective, the authors have addressed all other concerns very well.
> >
> > I really appreciate the effort the authors put into the rebuttal, and I now consider the paper to be a solid contribution. I am willing to raise my score to 8. I only ask that the authors incorporate carefully the clarifications and additional results from the rebuttal into the revised manuscript, because this will significantly strengthen the paper and benefit the whole tabular data watermark community.

---

> > > ### Author Response · Authors · 2025-11-25
> > >
> > > We sincerely thank the reviewer for the time and effort devoted to evaluating our work, as well as the thoughtful and constructive feedback, which has greatly improved the clarity and quality of the paper. We will carefully incorporate the clarifications and additional results from the rebuttal into the revised manuscript.

---

> ### Comment · Reviewer_6Ns1 · 2025-11-25
>
> Thank you for the rebuttal. The additional step is a good idea and I do believe it can addresses my concern on adding small noise to numerical datasets. I believe this significantly enhances the robustness of your method.
>
> Your response R8 is also satisfactory.

---

### Official Review · Reviewer_UNgJ · 2025-11-01

**Soundness:** 3
**Presentation:** 3
**Contribution:** 3
**Rating:** 6
**Confidence:** 4

**Summary:**

The paper introduces a novel approach to watermarking synthetic tabular data agnostic to the underlying generative model while maintaining high fidelity and robustness to attacks. Experiments are shown on a wide variety of datasets and generators to showcase the efficacy of the approach.

**Strengths:**

(a) The watermarking approach is agnostic to the generative model and simply uses multiple samples and picks the highest scoring one, with the scoring function appropriately chosen.
(b) Theoretical results are shown for detectability for a certain false positive rate of the watermarking approach and shown to be between 2 and 4 for a couple hundred samples@0.01.
(c) Watermarking both at a single (few) column-level and full set of columns are provided trading off robustness versus distortion trade-off.
(d) The computation time of generation and detection is better than the previous state-of-the-art approaches.

**Weaknesses:**

(i) The quantile rank is proposed as a method to thwart adversaries but it is unclear if this can not be reverse engineered by adversaries.
(ii) Similarly, the question of breaking the current approach for watermarking is not fully addressed in the paper.
(iii) The complexity of having both categorical and continuous features in the dataset is not discussed in detail.

**Questions:**

(1) If the dataset is complete categorical and say binary, would it pose a problem to the approach? More generally, can this be applied to video or image watermarking and what differs?
(2) The quantile rank is proposed but is deterministic which may make it vulnerable to attackers. Would selection by randomizing according to the rank help?
(3) How does the approach  work with respect to permutation attacks where the column order is completely randomized?

---

> ### Author Response · Authors · 2025-11-21
> **Response to Reviewer UNgJ (Part I)**
>
> We appreciate your valuable suggestions. We believe your concerns can be addressed through clarification and additional details in our responses below.
>
> ## About reverse-engineer the quantile rank
> >The quantile rank is proposed as a method to thwart adversaries but it is unclear if this can not be reverse engineered by adversaries.
>
> Thank you for raising this important concern. Below we clarify **(1)** how we instantiate “reverse-engineering” in our threat model and why recovering the quantile-rank configuration belongs to the harder class of attacks, and **(2)** empirical evidence showing that both types of reverse-engineering attacks are extremely difficult in practice.
>
> 1. **Two Levels of Reverse-Engineering Difficulty**
> There are two different goals an attacker may pursue:
>     - **Spoofing Attack (Easier)**: The adversary attempts to build a model to approximate the watermarked data distribution $P_{wm}$ to generate new valid watermarked samples, without necessarily recovering the exact secret key.
>     - **Parameter Recovery (Harder)**: The adversary attempts to deduce the secret parameters of the watermarking scheme—specifically the secret key $k$ or the exact configuration of the quantile rank (e.g., determining exactly which columns and quantile levels are being hashed). Success here constitutes a **total break**,: the attacker know both the watermark mechanism (under standard Kerckhoffs’ principle) and the secret key, then the attacker can perfectly scrub or spoof the watermark at will.
>
>
>     **Why Parameter Recovery is Harder**: If an attacker could perform Parameter Recovery, they could trivially perform Spoofing by re-using the recovered parameters. The reverse is not true: one may approximate a distribution statistically (Spoofing) without solving the cryptographic puzzle of recovering the underlying secret (Parameter Recovery).
>
> 2. **Reverse-engineering the quantile ranks is a hard Parameter-Recovery attack**
> Our paper describes JV hashing using the minimum, median, and maximum quantile ranks. However, in practice (and in standard cryptographic design), a simple extra step can make the quantile rank cryptographically secure. Specifically, for each row:
>    - We apply a keyed pseudorandom permutation to shuffle the column indices: $x\rightarrow \pi_k (x)$.
>    - After the permutation, we take the minimum, median, and maximum quantile ranks in that shuffled order.
>
>    Thus, an attacker would need to invert a pseudorandom permutation to get the actual quantile rank, which is equivalent to inverting the cryptographic hash.
>
> 3. **Empirical Evidence: even the easier Spoofing attack fails**: Our Adaptive Adversary experiments (Section 5.3) simulate a distillation attack, which represents the easier "Spoofing" threat. As shown in Figure 4, a powerful generative model (TabSyn) failed to learn the watermark distribution (AUC $\approx 0.5$). Since the adversary failed at the easier task of Spoofing (approximating the signal), it implies they statistically cannot succeed at the strictly harder task of Parameter Recovery (deducing the specific quantile rank configuration).
>
> **Update to the manuscript:**
> We add more discussion on watermark security in Sec 3.2.2 and include the full discussion in Appendix E.2.
>
> ## About breaking the watermark approach
> >Similarly, the question of breaking the current approach for watermarking is not fully addressed in the paper.
>
> This concern aligns closely with the reverse-engineering discussion above. Our analysis and experiments together show that the method is hard to break: the key-dependent quantile-rank configuration makes parameter recovery infeasible, and the adaptive-adversary experiment demonstrates that even the easier spoofing attack fails.
>
>
> ## About randomizing the quantile rank selection
> >The quantile rank is proposed but is deterministic which may make it vulnerable to attackers. Would selection by randomizing according to the rank help?
>
> We thank the reviewer for this suggestion. As detailed in our previous response, we have demonstrated that applying a pseudorandom permutation safeguard makes the quantile rank computationally difficult for attackers to estimate or steal. However, we acknowledge that our experiments rely on a general-purpose distillation framework, and a deterministic scheme may theoretically remain vulnerable to bespoke, optimized attacks.
>
> While adding randomness could improve robustness, it introduces a significant trade-off. A straightforward approach would be to leverage a set of secret keys (e.g., drawing a key uniformly at random for each instance [1]). However, this can severely impact detection performance (increasing FPR) and efficiency when the number of samples to watermark is large (which is common for tables), as the detector would need to test against every key in the set.
>
> **References**
>
> [1] Kuditipudi, Rohith, et al. "Robust distortion-free watermarks for language models." arXiv preprint arXiv:2307.15593 (2023).

---

> ### Author Response · Authors · 2025-11-21
> **Response to Reviewer UNgJ (Part II)**
>
> ## About complexity of mixed-type dataset
> >The complexity of having both categorical and continuous features in the dataset is not discussed in detail.
>
> >If the dataset is complete categorical and say binary, would it pose a problem to the approach?
>
> Thank you for the thoughtful questions.
> - Our method **does not depend on the data type (continous, discrete or mixed-type) itself**. This is due to the fact that our watermark method operates on the output sample of the tabular generative model, which is responsible to handle the complexity of the mixed-type data (e.g. map both continous and dicrite variable to continous latent in TabSyn).
> - Our method **relies on the standard assumption that the underlying unwatermarked data distribution has sufficient entropy**. In contrast, if the dataset is extremely low-entropy (e.g., fully binary), our method becomes fundamentally limited. With only two possible values in a column, repeated multi-sample selection would repeatedly choose the value that yields the higher score, which introduces noticeable distortion. Conversely, as implied by Theorem 4.3, maintaining the distortion-free property requires avoiding repeated usage of the same value, so each binary column can contribute at most two distinct watermarked samples. This yields too few watermarked rows and therefore weak detectability.
>
>   We note that the **requirement for sufficient entropy is standard across statistical watermarking methods** [1, 2, 3, 4], which demonstrate that adequate entropy is a prerequisite for embedding detectable signals without incurring significant distortion.
>
>
> [1] Kirchenbauer, John, et al. "A watermark for large language models." International Conference on Machine Learning. PMLR, 2023.
>
> [2] Zhao, Xuandong, et al. "Provable robust watermarking for ai-generated text." International Conference on Learning Representations 2024.
>
> [3] Kuditipudi, Rohith, et al. "Robust distortion-free watermarks for language models." Transactions on Machine Learning Research 2024.
>
> [4] Kirchenbauer, John, et al. "On the reliability of watermarks for large language models." International Conference on Learning Representations 2024.
>
>
> ## About application to image/video watermark
> >More generally, can this be applied to video or image watermarking and what differs?
>
> Our answer is two-fold: While MUSE is theoretically applicable to image and video generation, it is not suitable in practice due to different application setting. The standard scenario for tabular watermarking involves watermarking a table, which consists of a batch of i.i.d. samples, each row contributes positively to the aggregate $z$-score, thereby enhancing detectability. In contrast, image watermarking typically requires detection on a single instance. As shown in Theorem 4.1, for a target detactability of FPR=0.01%, MUSE requires a batch size of $N = 100$ and sampling repetitions $m = 4$ to achieve reliable detectability (i.e. 400 number of image generative model inference is required to watermark a batch of 100 images). This makes it prohibitive and not suitable for standard single-image watermark tasks. However, for specific real-world applications that involve generating and verifying batches of images, MUSE remains a potentially useful approach.
>
> **Update to the manuscript:**
> We have added a discussion on the difference between image and tabular watermark in Appendix F.5.

---

> ### Author Response · Authors · 2025-11-21
> **Response to Reviewer UNgJ (Part III)**
>
> ## About column permutation attack
> >How does the approach work with respect to permutation attacks where the columnl order is completely randomized?
>
> Thank you for raising this important point. In the following we will show 1) MUSE-PC is naturally robust to column permutation and 2) MUSE-JV can be made permutation-robust by estimating per-column statistics from synthetic samples and provide empirical evidence to support our claim.
>
> **PC Hashing.**
> PC hashing is inherently invariant to column order because a row’s score is computed as the sum over all columns, without relying on their positions. Therefore, a column permutation has no effect on its watermark detection.
>
>
> **JV hashing.**
> As presented in the paper, JV hashing assumes access to the correct per-column statistics (min/max) to compute quantile ranks; therefore, a full column permutation breaks this assumption. However, this limitation is not inherent to the method. A practical modification is to **estimate the min/max values directly from the synthetic table itself**, rather than loading pre-stored statistics. This makes the detector independent of column identities and immune to column permutation. The trade-off is that the estimated min/max may slightly deviate from the injector’s values and thus introduce mild score mismatch, but this affects detection only if the estimation is highly inaccurate.
>
> To evaluate this idea, we performed an experiment where both watermark injection and detection used min/max estimates computed from 10,000 independently generated samples. The results show that this estimation introduces negligible degradation in detectability:
>
> | Dataset | AUC   | T@F 0.1  |
> |---------|-------|----------|
> | Adult   | 1.000 | 1.000    |
> | Default | 0.997 | 0.809    |
> | Shoppers| 1.000 | 1.000    |
> | Beijing | 1.000 | 1.000    |
>
>
> **Update to the manuscript:**
> We have added a discussion on column permutation attack in Appendix E.1.

---

### Official Review · Reviewer_yLPz · 2025-11-04

**Soundness:** 3
**Presentation:** 2
**Contribution:** 3
**Rating:** 6
**Confidence:** 4

**Summary:**

The paper proposes MUSE, a model-agnostic watermarking framework for tabular generative models that eliminates the need for fragile DDIM inversion used in prior methods such as TabWak[1]. MUSE performs multi-sample selection. It samples multiple candidates ad selects the one with the highest key-dependent watermark score. Two watermarking scoring functions are introduced, one is JV Hashing, which hashes for low-distortion fidelity, the other is PC hashing, used for strong robustness. Theoretical analysis provides calibration of the false positive rate and provable distribution preservation.


[1] Zhu, C., Tang, J., Galjaard, J. M., Chen, P. Y., Birke, R., Bos, C., & Chen, L. Y. (2025, January). TabWak: A Watermark for Tabular Diffusion Models. In ICLR.

**Strengths:**

1. The approach departs from the inversion-based paradigm dominant in diffusion watermarking, meaning MUSE is compatible with any generative model that supports repeated sampling, including diffusion, autoregressive, and masked models.
2. Theoretical analysis on the detectability and distribution-preservation is provided. Though I didn't take a thorough look to the proof.

**Weaknesses:**

1. The choice of pseudorandom function $f$ and hash function $H$ is abstracted away but critical to real-world detectability and security. The paper doesn't provide implementation sensitivity analyses, e.g., whether certain hash or key spaces degrade performance.
2. No ablation on hyperparameters $m$ beyond theory. While theorem 4.1 gives calibration, the experiments mostly fix $m=2$. There is no systematic analysis of how varying $m$ or $N$ affects trade-offs between computation, detectability, and fidelity.
3. The paper demonstrates quantitative performance but provides little visualization of what kind of statistical signal the watermark introduces in the tabular domain.

**Questions:**

1. The field of tabular data watermarking is relatively new. Some baselines (Tree-Ring, Gaussian Shading) are from image watermarking. Could the author explain how these two methods are used in the tabular diffusion model?
2. The detector relies on a simple mean-score threshold; it’s unclear how well this scales to more complex scenarios (e.g., mixed distributions, partial key leakage, or federated settings).

---

> ### Author Response · Authors · 2025-11-21
> **Response to Reviewer yLPz (Part I)**
>
> We appreciate your valuable suggestions. We believe your concerns can be addressed through clarification and additional details in our responses below.
>
> ## About choice of hash function and key
> >The choice of pseudorandom function and hash function is abstracted away but critical to real-world detectability and security. The paper doesn't provide implementation sensitivity analyses, e.g., whether certain hash or key spaces degrade performance.
>
> We agree that the manuscript, while including an ablation on PRF output distributions (Bernoulli vs. Uniform), did not report a systematic sensitivity analysis on (1) the hash function family and (2) the secret key space. To address this, we conducted a new set of experiments measuring the robustness of MUSE under a variety of hash families and keys of varying entropy.
>
> ---
>
> 1. **Sensitivity Analysis: Hash Function Families**
> We replace the cryptographic hash functions with a different family of available ones in the Python hashlib library:
>
>     - SHA-256 (default one used in the paper)
>     - SHA3-256
>     - BLAKE2s
>     - SHAKE-256
>     - MD5
>
>     For each hash family, we repeat the experiment 13 times using randomly sampled keys with bit-lengths $[32, 40, 48, 56, 64, 72, 80, 88, 96, 104, 112, 120, 128]$.
>
>     We evaluate detectability (ROC-AUC and TPR@0.1%FPR) and fidelity (Marginal, Correlation, C2ST and MLE) on the Adult dataset with $N = 500, m = 2$, keeping the PRF mapping to Bernoulli(0.5) fixed. We report the average value and the standard deviation computed across different private keys.
>
>     **MUSE-PC**
>     | **Hash**   | **Marg.**        | **Corr.**        | **C2ST**        | **MLE**         | **AUC** | **TPR@0.1%FPR** |
>     |------------|------------------|------------------|------------------|------------------|--------|------------------|
>     | sha256     | 0.947±0.009      | 0.92±0.011       | 0.772±0.023      | 0.907±0.001      | 1      | 1                |
>     | sha3_256   | 0.945±0.009      | 0.919±0.011      | 0.776±0.023      | 0.907±0.002      | 1      | 1                |
>     | blake2s    | 0.95±0.007       | 0.925±0.008      | 0.78±0.018       | 0.907±0.002      | 1      | 1                |
>     | shake_256  | 0.95±0.007       | 0.925±0.009      | 0.785±0.021      | 0.908±0.001      | 1      | 1                |
>     | md5        | 0.943±0.011      | 0.915±0.015      | 0.767±0.020      | 0.907±0.002      | 1      | 1                |
>
>
>    **MUSE-JV**
>
>     | **Hash**   | **Marg.**        | **Corr.**        | **C2ST**        | **MLE**         | **AUC** | **TPR@0.1%FPR** |
>     |------------|------------------|------------------|------------------|------------------|--------|------------------|
>     | sha256     | 0.977±0.004      | 0.960±0.005      | 0.916±0.016      | 0.909±0.001      | 1      | 1                |
>     | sha3_256   | 0.977±0.006      | 0.961±0.008      | 0.913±0.024      | 0.908±0.002      | 1      | 1                |
>     | blake2s    | 0.974±0.005      | 0.956±0.006      | 0.893±0.024      | 0.909±0.002      | 1      | 1                |
>     | shake_256  | 0.977±0.005      | 0.961±0.006      | 0.915±0.010      | 0.909±0.001      | 1      | 1                |
>     | md5        | 0.975±0.006      | 0.958±0.007      | 0.912±0.022      | 0.909±0.001      | 1      | 1                |
>
>
>     From these results (also see figure 6 in the updated submssion), we observe that detectability remains perfect (AUC = 1, TPR@0.1%FPR = 1) for all hash families, including weaker ones such as MD5, and that fidelity metrics (Marg., Corr., C2ST, MLE) stay consistently stable with all variations falling well within statistical noise. This shows that **MUSE is insensitive to the specific choice of hash function, and its performance is robust across a wide range of hash families**.
>
> 2. **Sensitivity Analysis: Secret Key Space**
>   Recall that the standard deviations shown in the above table are computed across multiple runs with independently sampled secret keys with bit-lengths $[32, 40, 48, 56, 64, 72, 80, 88, 96, 104, 112, 120, 128]$. We can see the variance is at a very small scale, which indicates that the choice of key has a negligible impact on both detectability and fidelity, confirming that MUSE is insensitive to the specific key used.
>
>
> **Update to the manuscript:**
> We have added detailed sensitivity analysis experiments on the hash function and key space in Section 5.4 and Figure 6.
> This section reports empirical results across
> (i) five hash families (SHA-256, SHA3-256, BLAKE2s, SHAKE-256, MD5), and
> (ii) key lengths from 32 to 128 bits.

---

> ### Author Response · Authors · 2025-11-21
> **Response to Reviewer yLPz (Part II)**
>
> ## About ablation on repeated number $m$
> >No ablation on hyperparameters m beyond theory. While Theorem 4.1 gives calibration, the experiments mostly fix m=2. There is no systematic analysis of how varying m or N affects trade-offs between computation, detectability, and fidelity.
>
> We thank the reviewer for pointing out this missing analysis.
>
> **How m affects detectability and fidelity tradeoff**
> We set $N=100$ and sweep $m\in [2, 4, 6, 8, 10, 12, 14, 16]$ on all four datasets (Adult, Default, Shoppers, Beijing) under both JV hashing and PC hashing. For each setting, we draw a scatter plot where the x-axis represents the data fidelity, computed by averaging four data quality metrics (Marginal, Correlation, C2ST, and MLE). While the y-axis shows the $z$-statistic.
>
> From the figure, we observe that
> 1) For both JV and PC hashing, increasing $m$ leads to better detectability and worse data fidelity, confirming the tradeoff between detectability and data fidelity.
> 2) Under the same $m$, PC hashing incurs better detectability compared to JV (except for the Shoppers dataset under $m=2,4$), while getting worse data fidelity. This validates our design principle of these two hashing strategies.
>
>
> **How N affects detectability and fidelity tradeoff**
> We first note that the number of rows of the watermarked table $N$ is a user-determined parameter, but not a parameter of our model. The fidelity does not depend on $N$ since it is evaluated on a fixed number of rows (the same as the training dataset for fair comparison).
>
> Detectability does depend on $N$. Intuitively, since the watermark is embedded independently on each row, the larger the $N$, the larger the detectability since each row contributes to the $z$-score. To validate this, we set $m=2$ and sweep $N\in [100, 500, 1000, 1500, 2000, 2500, 3000, 3500]$ and plot the corresponding $z$-score. From the plot, we can observe that the detectability increases as N increases.
>
>
> **How m and N affect computation**
> The watermark generation time scales linearly with $m$ and $N$.
>
>
> **Update to the manuscript**
> We add an ablation study for the parameter $m$ and the number of watermarked row $N$ in Sec 5.2, Figure 4 and Figure 7.
>
> ## About visualization of watermarked signal
> >The paper demonstrates quantitative performance but provides little visualization of what kind of statistical signal the watermark introduces in the tabular domain.
>
> Thank you for pointing this out. We have added visualizations that directly illustrate the statistical signal introduced by our watermark in both the JV and PC hashing variants.
>
> **JV-hashing**
> For JV hashing, each row-level score is a PRF following Bernulli(0.5). We plot the empirical probability mass function (PMF) of these scores for both watermarked and unwatermarked tables. As expected, the unwatermarked data yields an approximately symmetric distribution over {0,1}. while watermarked tables exhibit a clear shift of probability mass toward larger score values due to multi-sample selection.
>
> **PC-hashing**
> For PC hashing, the row-level score is the sum of per-column Bernoulli bits, taking values in {0,...n} where n is the number of columns. We visualize the empirical PMF over the normalized score (defined in Equation 3 in paper). Again, unwatermarked tables show the expected symmetric distribution, while watermarked tables exhibit a rightward shift in mass, reflecting a higher expected number of matched column bits.
>
> **Update to the manuscript:**
> We have added a new section in Appendix C.3 to include the visualization plots.

---

> ### Author Response · Authors · 2025-11-21
> **Response to Reviewer yLPz (Part III)**
>
> ## About image watermark baselines
> >The field of tabular data watermarking is relatively new. Some baselines (Tree-Ring, Gaussian Shading) are from image watermarking. Could the author explain how these two methods are used in the tabular diffusion model?
>
> We follow the methodology and adaptation established in TabWak [1]:
>
> - **Tree-Ring Watermark**: We treat the full table ($m$ rows $\times$ $n$ columns) as a latent image. However, unlike square images, tables typically possess a high aspect ratio (significantly more rows than columns). To accommodate this tall, rectangular shape, we embed a ripple-shaped pattern across the latent space rather than the standard centralized ring pattern used in image tasks.
> - **Gaussian Shading**: Unlike Tree-Ring watermark, this method is applied on a per-row basis (analogous to treating each row as a image). We enforce a fixed control seed across all rows (since keeping track of a different seed for each row is challenging due to row shuffing attak). This ensures the watermark remains detectable even if the rows are permuted.
>
> For the full algorithmic details, please see Appendix D.1 and D.2 of TabWak [1].
>
> [1] Zhu, Chaoyi, et al. "Tabwak: A watermark for tabular diffusion models." International Conference on Learning Representations. ICLR 2025.
>
> **Update to the manuscript:**
> We have added a description of the image watermark baselines in Appendix F.5 for completeness.
>
> ## About more complex scenarios
> >The detector relies on a simple mean-score threshold; it’s unclear how well this scales to more complex scenarios (e.g., mixed distributions, partial key leakage, or federated settings).
>
> We appreciate the reviewer pointing out these scenarios. We believe the simplicity of the mean-score statistic is not a bottleneck for complex scenarios. We address each scenario below:
>
> 1. **Mixed Distributions**: We interpret this as scenarios where watermarked data is interleaved with clean data. Our "Row Deletion" robustness experiment (Sec. 5.3, Figure 5) directly simulates this by mixing unwatermarked samples into the table. Results show that MUSE-JV and MUSE-PC maintain high detection efficacy even when 60–80% of the distribution is "mixed" (i.e., replaced with unwatermarked data).
> 2. **Partial Key Leakage**: This is closely related to the security analysis detailed in our response to Reviewer UNgJ. We adhere to the standard security setting (Kerckhoffs's principle): the watermark mechanism is public, but the private key is not. Our adaptive adversary experiments (Sec. 5.3) demonstrate that it is computationally difficult for attackers to reverse-engineer the private key. We refer the reviewer to "About reverse-engineer the quantile rank" for full details.
> 3. **Federated Settings**: To the best of our knowledge, the primary focus of watermark in feterated learning in current literature is *model watermarking* [1,2] (e.g. to claim ownership of a trained ML model), however, our method belongs to *data watermarking* (i.e. watermark a table). Since federated learning keeps data local and private to each client, applying a data watermark is less relevant to the aggregation process; however, our method can be useful for local client-side verification if needed.
>
> [1] Atli, Buse Gul et al. “WAFFLE: Watermarking in Federated Learning.” 2021 40th International Symposium on Reliable Distributed Systems (SRDS) (2020): 310-320.
>
> [2] Yang, Wenyuan, et al. "Watermarking in secure federated learning: A verification framework based on client-side backdooring." ACM Transactions on Intelligent Systems and Technology 15.1 (2023): 1-25.

---

### Comment · Area_Chair_xeS1 · 2025-11-25
**Authors' responses**

Dear Reviewers,

The authors have submitted their responses to your questions and feedbacks. Please read them and give your comments.

Regards,
AC

---

### Comment · Area_Chair_xeS1 · 2025-11-28
**The Author/Reviewer Discussion Phase deadline is approaching.**

Dear Reviewers,

The Author/Reviewer Discussion Phase deadline is approaching. If you have not responded to authors’ rebuttal, please read it and give your feedback asap.

Regards,
AC

---

### Author Response · Authors · 2025-12-01
**Summary of Rebuttal and Revisions**

We want to provide the following summary of revisions to help with the final evaluation.

We highlight that Reviewer **6Ns1** engaged deeply with the submission and, following our rebuttal, **raised their score from 2 to 8 (prior to the November 27th incident), leading to improved scores of 6, 6, 8 (from 6, 6, 2)**.  They stated that the paper is now a "solid contribution" and that their concerns were addressed "very well."
- About Novelty: We clarified the novelty of our work by: 1) highlighting the fundamental structural differences between tabular and LLM watermarking (using KGW [1] and Unigram [2] as examples); and 2) distinguishing our approach from related works Watermax [3] and Bahri & Wieting (2025) [4]. A detailed discussion has been added to the Related Work section.
- About Robustness: We included an additional attack method: Gaussian additive noise (Section 5.3). Notably, we addressed concerns regarding global perturbations on numerical datasets by introducing a quantization preprocessing step, which we validated through new experiments on two new datasets: California and Letter (Remark 3.1 and Appendix E.2).
- About Detection Framework: To address inaccurate statistical estimation under the null hypothesis in low-entropy scenarios, we extended the original detection framework derivation with a Monte Carlo estimation method (Appendix F.4).

We also incorporated constructive feedback from Reviewers **yLPz** and **UNgJ**. We added comprehensive sensitivity analyses on hash functions and key spaces (Section 5.4) and ablation studies on hyperparameters $m$ and $N$ (Section 5.2). We added a discussion distinguishing tabular watermarking from image/video watermarking in Appendix F.5. Additionally, we visualized the watermark signal (Appendix C.3), expanded the security analysis regarding reverse-engineering (Remark 3.2 and Appendix E.3), and discussed robustness against column permutation (Appendix E.1).

We believe the discussion phase has significantly strengthened the paper. We sincerely thank the reviewers for their constructive insights and are confident that the revised manuscript addresses all major concerns raised during the initial review.

[1] Kirchenbauer, John, et al. "A watermark for large language models." International Conference on Machine Learning. PMLR, 2023.

[2] Zhao, Xuandong, et al. "Provable robust watermarking for ai-generated text." ICLR 2024.

[3] Giboulot, Eva, and Teddy Furon. "WaterMax: breaking the LLM watermark detectability-robustness-quality trade-off." Advances in Neural Information Processing Systems 37 (2024): 18848-18881.

[4] Bahri, Dara, and John Wieting. "A watermark for black-box language models." arXiv preprint arXiv:2410.02099 (2024).

---

### Meta-Review · Area_Chair_7wU4 · 2026-01-09

**Summary:**

The paper proposes a new watermarking approach for tabular diffusion models: sampling several candidates and selecting one according to a scoring function
Main concerns:
Reviewer 6Ns1 (initial score 2):  The idea has been proposed before;
Reviewer yLPz (initial score 6): No analysis on wether the scoring function and hash space degrades the performance (sensitivity analysis); no hyperparameter analysis.
Reviewer UNgJ (initial score 6): Not clear, if the proposed approach can not be used by adversaries.

**Reviewer Concerns:**

The rebuttal has been done (very detailed) which addressed the reviewers concern. The 'negative' reviewer actually agreed to raise the score before the incident from 2 to 8 (!). Two others did not comment.

**Reviewer Scores:**

They will be 8, 6, 6.

---

### Decision · Program_Chairs · 2026-01-26

Accept (Poster)